



# Modelling rainfall with a Bartlett-Lewis process: New developments

Christian Onof[1] and Li-Pen Wang[1]

[1]Department of Civil and Environmental Engineering, Imperial College London, SW7 2AZ, UK

**Correspondence:** Christian Onof (c.onof@imperial.ac.uk)

**Abstract.** The use of Poisson-cluster processes to model rainfall time series at a range of scales now has a history of more than 30 years. Among them, the Randomised (also called modified) Bartlett-Lewis model (RBL1) is particularly popular, while a refinement of this model was proposed recently (RBL2) (Kaczmarska et al., 2014). Fitting such models essentially relies upon minimising the difference between theoretical statistics of the rainfall signal and their observed estimates. The first are obtained

using closed form analytical expressions for statistics of order 1 to 3 of the rainfall depths, as well as useful approximations of the wet-dry structure properties. The second are standard estimates of these statistics for each month of the data. This paper discusses two issues that are important for optimal model fitting of the RBL1 and RBL2. The first is that, when revisiting the derivation of the analytical expressions for the rainfall depth moments, it appears that the space of possible parameters is wider than has been assumed in the past papers. The second is that care must be exerted in the way monthly statistics are estimated

from the data. The impact of these two issues upon both models, in particular upon the estimation of extreme rainfall depths at hourly and sub-hourly timescales is examined using 69 years of 5-min and 105 years of 10-min rainfall data from Bochum (Germany) and Uccle (Belgium), respectively.

## 1 Background

The objective of stochastic rainfall modelling is to provide tools that enable the generation of long realistic series of rainfall.

These can then be used as inputs to catchment hydrological models, to erosion models, to sewerage discharge models, or even be used to examine the frequency of outages in telecommunications networks (Connolly et al., 1998; Arnbjerg-Nielsen, 2012; Arnbjerg-Nielsen et al., 2013; Onof and Arnbjerg-Nielsen, 2009; Wang et al., 2010). Depending upon the application, 'realistic' will mean different things. For applications that are related to design, 'realistic' will involve the reproduction of the observed extreme behaviour of the precipitation process at a range of scales.

In this paper, we focus upon one particular approach to rainfall modelling, that which is based upon the use of point processes as defining the times at which the building blocks of the model, i.e. rainfall cells, arrive. These cells are conceptual ones, although their typical characteristics are those of Small Mesoscale Areas (SMSA) which are embedded in Large Mesoscale Areas (Burlando and Rosso, 1993). The presence of clustering means that a homogeneous Poisson point process is not an appropriate choice for the underlying process of cell arrivals. Two options are available. The first introduces randomness by

having the Poisson rates behave as a continuous-time Markov chain: this defines a Cox (doubly-stochastic) process (see Ramesh (1995); Ramesh et al. (2018)).





The second explicitly models the clustering process. This can be done by defining the number cells in a storm as a random variable, with another random variable modelling the delays from the storm to the cell arrival time. This defines a Neyman-Scott process (see Cowpertwait (1998); Evin and Favre (2008); Paschalis et al. (2014)). Alternatively, a second homogeneous

Poisson process defines the cell arrival times over a duration of storm activity that defines a random variable. This defines a Bartlett-Lewis process (see Onof and Wheater (1993); Khaliq and Cunnane (1996); Verhoest et al. (1997); Kossieris et al. (2018)). For both Poisson-cluster processes, the SMSAs are then represented by rectangular pulses corresponding to a random constant rainfall intensity over a random duration. In this paper, the Bartlett-Lewis process is the chosen point process model.

Two issues have been flagged in the literature which limit the applicability of a number of variants of the basic model type

published in 1987 (Rodriguez-Iturbe et al., 1987). The first one is well-known (e.g. Verhoest et al. (2010)). Many studies have shown that Rectangular Pulse models underestimate hourly (and sub-hourly) extremes (Verhoest et al. (2010) and references therein). This is often accompanied by an overestimation of daily extremes. The other less well-known problem was identified by Marani (2003). While one of the strengths of models based upon Poisson-cluster processes is their ability to capture rainfall variability over a range of scales (hence its use in disaggregation - see Koutsoyiannis and Mamassis (2001); Koutsoyiannis

et al. (2003); Kossieris et al. (2016)), they underestimate this variability for scales equal to or larger than a few days.

Both issues are closely connected to fundamental features of these models and of the way they are fitted. The first arises partly due to the fact that the model is calibrated in such a way as to reproduce the mean behaviour of the precipitation process. That is, statistics like the mean, variance, autocovariance of rainfall totals at time-scales varying from one to 24 hours are used to fit the models. As far as the cell intensity parameters are concerned, these statistics are functions of their first and second-

45 order moments only. The rest of this distribution is not thereby determined, although the choice of distribution has a clear impact upon the extremes (Onof and Wheater, 1994). This situation can be partially addressed by including the coefficients of skewness (hereafter, 'skewness') of the rainfall depths at relevant time-scales as additional statistics in the calibration of the model (Cowpertwait, 1998). Kaczmarska et al. (2014) similarly find that the inclusion of the skewness yields a reasonable performance and extend the range of time-scales to include sub-hourly scales which are of key importance in urban hydrology,

erosion studies and telecommunications applications. There remains however the option of using a fat-tailed distribution for the cell intensity to achieve further improvement. To see whether this is advisable/useful, we need to get a better picture of what produces the extremes at different time-scales: is it predominantly the superposition of several cells, or is it mostly the rainfall produced by a single cell. In the latter case, the choice of a different distribution of rainfall intensities is a key decision.

The other issue had not so far received much attention although it is in fact of clear practical import. If we want the model to

55 be able to capture longer term variability (as would certainly be required to reproduce climate variability for instance), then this issue must be addressed. The most promising ways forward in this respect come from combining the Poisson-cluster model with a coarse-scale model that captures much of the longer-term variability (Park et al., 2019), or from letting climatological information guide the weighting to be assigned to different months in the data in calibrating the model (Kaczmarska et al. (2015); Cross et al. (2019)). Both approaches represent important developments. The first approach involving the combination

of two models has the advantage of enabling a much improved reproduction of extreme rainfall depths. The second approach which incorporates climatological information, enables this model to be used as weather generator in climate impact studies.





While the use of extraneous (e.g. climatological) information and the combination with another (e.g. coarse-scale) model are the most promising ways in which this area of stochastic rainfall modelling is developing, the issue of how the Poisson-cluster model is fitted to rainfall statistics needs to be revisited. In this paper, we address two hitherto unnoticed issues with random
parameter Bartlett-Lewis rainfall models. First, we draw attention to a claim made in the original publication of the randomised Bartlett-Lewis model (Rodriguez-Iturbe et al., 1988) which involves an erroneous assessment of the mathematically feasible limits of a key model parameter. Correcting this misspecification of the constraints on this parameter allows us to consider a broader parameter space, thereby potentially including parameter values that will improve model performance. Second, we show the importance of the choice of estimators for the statistics used in model fits to individual months. We shall show that,
by taking both issues into account, it is possible to improve the reproduction of extreme rainfall depths over a range of scales. A detailed examination of the impact upon the variance function will be carried out in another paper.

This paper starts with a presentation of the data and a reminder of the structure of three versions of the Rectangular Pulse Bartlett-Lewis model, as well as of how these models are fitted. The revised equations for the statistics of order 1 to 3 of the rainfall depths at aggregation scale $h$ hours are then presented. In the following section, we discuss the estimation of standard
monthly statistics, and show the bias that can be introduced through the use of a commonly used type of estimator. In the final section, we consider the impact of the new equations and unbiased estimation method upon the reproduction of standard statistics and extremes of rainfall depths.

## 2   Data

Five-minute rain gauge rainfall data from a rain gauge in Germany (North-Rhine-Westphalia) and one in Belgium (Flanders:
Brussels region) are used to demonstrate the new developments in model (population) and data (sample) statistics for model fitting described in this paper. These are:

– Bochum: 69 years of 5 minute data from January 1981 to December 1999;

– Uccle: 105 years of 10 minute data from January 1898 to December 2002.

Because of constraints on the length of the paper, the results for Uccle are shown only in the Supplement of this paper.
Additionally, for the purpose of the numerical investigation of estimators, Greenwich (14.5 years of 5 minute data from February 1987 to July 2001) data are used.

## 3   Model structure

In the original Bartlett-Lewis Rectangular Pulse (OBL) model (as illustrated in Fig. 1), storms arrive according to a Poisson process at rate $\lambda$. Another process generates cells associated with each storm: this is also a Poisson process, triggered by
90 the storm arrival (rate $\beta$), and active over a duration that is exponentially distributed with parameter $\gamma$. These cells have an exponential duration (parameter $\eta$) and a random depth (described by its first three (non-centred) moments: $\mu_x$ ; $\mu_{x2}$; $\mu_{x3}$).





Further development of the original model proposed by Rodriguez-Iturbe et al. (1987) involved has in particular focused upon the randomisation of the temporal structure of storms for the Bartlett-Lewis process (Rodriguez-Iturbe et al., 1988; Onof and Wheater, 1993). The temporal structure of precipitation is allowed to vary from storm to storm by randomising parameter

$\eta$. This can be chosen as a gamma distribution $\Gamma(\alpha, 1/\nu)$ distributed random variable that varies between storms. The cell arrival rate and storm duration parameter are scaled accordingly: $\beta = \kappa\eta$; $\gamma = \phi\eta$. This will be referred to as the Randomised Bartlett-Lewis model version 1 (RBL1).

Recently, this randomisation strategy was extended to include all the parameters describing the internal structure of the storm, i.e. to include parameter $\mu_x$ (Kaczmarska et al., 2014). $\mu_x$ is now a random variable that takes on different values for

different storms, proportionally to $\eta$: $\mu_x = \iota\eta$. This is the Randomised Bartlett-Lewis model version 2 (RBL2). This model was shown to outperform the OBL and RBL1 by Kaczmarska et al. (2014), but:

- only one data set was examined in that study so this conclusion cannot be generalised;

- the RBL1 was excluded from the comparison because the authors 'concluded that the improvement in the fit to proportion dry that had previously been found by randomizing $\eta$ was at the expense of a deterioration in the fit to the skewness'

(ibid.); but given the popularity and successful application of this model to a range of types of rainfall (e.g. see Onof et al. (2000)), we decided to include it here for further analysis.

## 4   Model calibration and the revised equations

### 4.1   Calibration

The OBL, RBL1 and RBL2 models generate rainfall as a continuous-time process, $\{Y(t)\}_{t\in\mathbb{R}}$: $Y(t)$ is the continuous-time

rainfall intensity at time $t$ resulting from the superposition of the intensities of all the cells active at time t. Rainfall records are, however, available in aggregated form for discrete time-scales. The rainfall depth $Y_i^{(h)}$ for a level of aggregation $h$ hours is given by:

$$Y_i^{(h)} = \int\limits_{(i-1)h}^{ih} Y(t)\,dt \tag{1}$$

Analytical expressions of the moments of the aggregated process $Y_i^{(h)}$ have been derived as functions of the model parameters.

Expressions for other statistical descriptors such as the proportion of dry periods at time scale $h$, have also been derived (see Onof et al. (2000) and Kaczmarska et al. (2014)).

The models are calibrated by a Generalised Method of Moments. That is, the model parameters are chosen so that the model values calculated with the available analytical expressions are as close as possible to the empirical values of these statistics obtained from observed data. This is achieved by minimising an objective function:

$$\sum_{\mathcal{M}\in\Omega} \omega(\mathcal{M})\left\{\mathcal{M} - \hat{\mathcal{M}}\right\}^2 \tag{2}$$





where $\Omega$ is a set of statistical descriptors, $\omega(\mathcal{M})$ a weight assigned to that property in the objective function, and $\hat{\mathcal{M}}$ is the estimate of that property from the sample of available data. For details about the optimal choices of the weights, see Kaczmarska et al. (2014).

In this paper, following the best practise suggested in Kaczmarska et al. (2014), we choose mean 1-h rainfall depth, and coefficient of variation, autocorrelation lag-1 and coefficient of skewness at 5/10-min (5 min for Bochum, 10 min for Uccle), 1-, 6- and 24-hour time-scales as statistical descriptors for the model calibration. In addition, inspired by the optimisation method proposed in Efstratiadis et al. (2002), we used the Simulated Annealing algorithm to search a promising region, and then the downhill simplex Nelder-Mead algorithm to identify the optimum to minimise Eq. (2).

Below, we present the methodology used to derive the new equations for the two randomised versions of the Bartlett-Lewis model.

## 4.2 Derivation of the new equations

As explained in Rodriguez-Iturbe et al. (1988), the mean and variance of the RBL1 - and this also applies to the RBL2 - are obtained by taking means over $\eta$ of these moments for the OBL. This is the case because the expressions of these moments only contain terms corresponding to contributions from single storms, i.e. $\lambda^q \eta^p$ terms with $q = 1$ only, as can be seen from the equations obtained by Rodriguez-Iturbe et al. (1987). The same goes for the derivation of the third-order centred moment.

In this section, we focus upon the derivation of the variance of the RBL1. The complete sets of new equations for RBL1 and RBL2 are presented in Appendix A.

The starting point for the derivation is the equation for the variance of the OBL model. Here, rather than use the original OBL model parameters (Rodriguez-Iturbe et al., 1987), i.e.

$$\{\lambda, \gamma, \beta, \eta, \mu_x, \mu_{x^2}, \mu_{x^3}\}$$

We replace the second and third parameters by dimensionless parameters $\phi$ and $\kappa$ that are also used in RBL1 and RBL2 so that the parameterisation of the OBL is now in terms of:

$$\{\lambda, \phi, \kappa, \eta, \mu_x, \mu_{x^2}, \mu_{x^3}\}$$

where $\gamma = \phi\eta$ and $\beta = \kappa\eta$.

In the analytical expression for the OBL variance, we make the dependence upon parameter $\eta$ explicit by referring to it as $V(h, \eta)$, This distinguishes it from the corresponding variances for RBL models denoted $V(h)$. The OBL variance is:

$$
\begin{aligned}
V(h, \eta) &= \frac{2\lambda\mu_c}{\eta}\left[\frac{(\mu_{x^2} + \kappa\mu_x^2/\phi)h}{\eta} + \frac{\mu_x^2\kappa(1 - e^{-\phi\eta h})}{\phi^2\eta^2(\phi^2 - 1)} - \left(\mu_{x^2} + \frac{\kappa\phi\mu_x^2}{\phi^2 - 1}\right)\frac{1 - e^{-\eta h}}{\eta^2}\right] \\
&= 2\lambda\mu_c\mu_x^2\left\{\left(f_1 + \frac{\kappa}{\phi}\right)\frac{h}{\eta^2} + \left(\frac{\kappa}{\phi^2(\phi^2 - 1)}\right)\frac{1 - e^{-\phi\eta h}}{\eta^3}\right. \\
&\quad \left. + \left(f_1 + \frac{\kappa\phi}{\phi^2 - 1}\right)\frac{1 - e^{-\eta h}}{\eta^3}\right\}
\end{aligned}
\tag{3}
$$

where $f_1 = \mu_{X^2}/\mu_X^2$ and $f_2 = \mu_{X^3}/\mu_X^3$. For more on the choice of these parameters, see Appendix B.





When deriving the expression for a moment $\mathcal{M}$ in the RBL models, we multiply the corresponding moment $\mathcal{M}(\eta)$ for the OBL model by the density function $f$ of the Gamma distribution $\Gamma(\alpha, 1/\nu)$ of $\eta$ and integrate over all possible values of $\eta$:

$$\mathcal{M} = E_\eta\left[\mathcal{M}(\eta)\right] = \int_0^\infty \mathcal{M}(\eta) f(\eta)\, d\eta \tag{4}$$

where the density function of the Gamma distribution is given by:

$$f(\eta) = \frac{\eta^{\alpha-1} \nu^\alpha e^{-\nu\eta}}{\Gamma(\alpha)}\, d\eta \text{ if } \eta \geq 0$$
$$f(\eta) = 0 \text{ if } \eta < 0$$

The issue of the convergence of these integrals has, however, not been addressed explicitly in the literature (aside from a mention in Kaczmarska et al. (2014)). The integration involves integrals of the following general type evaluated at $l = 0$:

$$T(k, u, l) = \int_l^{+\infty} \eta^{-k} e^{-u\eta} \frac{\eta^{\alpha-1} \nu^\alpha e^{-\nu\eta}}{\Gamma(\alpha)}\, d\eta$$

$$= \frac{\nu^\alpha}{(\nu+u)^{\alpha-k}} \frac{\Gamma(\alpha-k, l(\nu+u))}{\Gamma(\alpha)}$$

$$\tag{5}$$

where $\Gamma(s)$ is the (complete) Gamma function, and $\Gamma(s, x)$ the incomplete Gamma function, defined as:

$$\Gamma(s, x) = \int_x^\infty t^{s-1} e^{-t}\, dt$$

When $l = 0$, the integration in (5) is possible (i.e. the integral is finite) if and only if the integrand is integrable in the neighbourhood of 0, since there are no problems of convergence at $\infty$. If we look at the terms the integrand comprises for the
statistics that are used to fit the model, we find that they behave in the neighbourhood of 0 as $\eta^{\alpha-n}$ with $n = 2$ for the mean rainfall intensity, $n \leq 4$ for its variance and covariance, and $n \leq 5$ for its third-order central moment. The integrals of such terms converge as long as $\alpha - n > -1$, i.e. $\alpha > n - 1..$

It therefore seems that, for the RBL1, $V(h)$ is finite as long as $\alpha > 3$. Similarly, as the expressions in Appendix A2 show, the
mean $M(h)$ is finite as long as $\alpha > 1$, the covariance of lag-$k$ $C(k, h)$ is finite when $\alpha > 3$ and the third-order centred moment $S(h)$, when $\alpha > 4$.

This conclusion is however too hasty. Indeed, it involves considering separately the integration of each additive term in Eq. (3). It is with such separate integration that the expressions for the variance and covariance used in Rodriguez-Iturbe et al. (1988) are obtained, and these expressions were used in subsequent research.

Insofar as only moments of order less than 3 were used in past studies (the third-order moment for the RBL1 was only published in a report (Onof et al., 2013) and therefore not used in most of the literature), the constraint $\alpha > 3$ applied to the fits found in these past papers (e.g.(Rodriguez-Iturbe et al., 1988), (Khaliq and Cunnane, 1996), (Verhoest et al., 1997), (Onof et al., 2000), (Verhoest et al., 2010) and (Kim et al., 2017)). However, since the issue of the convergence of these integrals was





not examined, it is not surprising to find, in most of these studies, that values of $\alpha$ below 3, i.e. outside the domain of feasibility

of the optimisation, are obtained for some months. The parameter sets for these months are thus not feasible and *a fortiori* not

optimal, contrary to the claims made in those papers.

We now need to check whether, when proceeding without separating the integration into the sum of integrals of the additive

terms in the integrand, the domain of convergence of the integral is still defined by $\alpha \in (3, +\infty)$ for the variance (and for

the covariance, and $\alpha \in (4, +\infty)$ for the third-order moment). That is, are any values of $\alpha$ for which the individuals integrals

diverge, but the integral of the whole integrand does not? That would be the case for instance if, in the neighbourhood of $0$, the

terms leading to a divergence for certain values of $\alpha$ were to cancel out.

In line with Eq. (4), the variance $V(h)$ of the RBL1 model is obtained as:

$$V(h) = \int_0^\infty V(h, \eta) f(\eta) \, d\eta \tag{6}$$

and if we choose a small value $\eta_0$ of $\eta$, this integral is the sum:

$$190 \quad V(h) = \int_0^{\eta_0} V(h, \eta) f(\eta) \, d\eta + \int_{\eta_0}^\infty V(h, \eta) f(\eta) \, d\eta \tag{7}$$

whereby only the first integral has a limited domain of convergence. Let us call this first term $V_1(h)$.

From Eq. (3), we have:

$$V_1(h) \quad = \quad \frac{2\lambda\mu_c\mu_x^2}{\Gamma(\alpha)} \int_0^{\eta_0} \left[ \eta^{\alpha-3}\nu^\alpha e^{-\nu\eta} \left( f_1 + \frac{\kappa}{\phi} \right) h + \eta^{\alpha-4}\nu^\alpha e^{-\nu\eta} \left( \frac{\kappa(1-\phi^3)}{\phi^2(\phi^2-1)} - f_1 \right) \right.$$

$$\left. - \eta^{\alpha-4}\nu^\alpha e^{-(\nu+\phi h)\eta} \left( \frac{\kappa}{\phi^2(\phi^2-1)} \right) + \eta^{\alpha-4}\nu^\alpha e^{-(\nu+h)\eta} \left( f_1 + \frac{\kappa\phi}{\phi^2-1} \right) \right] \, d\eta$$

By doing first and second-order Taylor expansions of the exponential terms, we find that the $\eta^{\alpha-4}$ and $\eta^{\alpha-3}$ terms cancel, so

that after some algebra, we get:

$$V_1(h) \quad = \quad \frac{2\lambda\mu_c\mu_x^2}{\Gamma(\alpha)} \int_0^{\eta_0} \left[ \eta^{\alpha-2}\nu^\alpha \frac{h^2}{2} \left( \frac{\kappa}{\phi+1} + f_1 \right) + o(\eta^{\alpha-2}) \right] \, d\eta$$

which yields:

$$V_1(h) \approx \frac{\lambda\mu_c\nu^\alpha h^2 \mu_x^2 \eta_0^{\alpha-1}}{(\alpha-1)\Gamma(\alpha)} \left( \frac{\kappa}{\phi+1} + f_1 \right) \tag{8}$$

as long as $\alpha - 2 > -1$, i.e. $\alpha > 1$. Else, $V_1(h)$ is infinite.





This second term $V_2(h)$ is thus calculated as:

$$
\begin{aligned}
V_2(h) &= \int_{\eta_0}^{\infty} V(h,\eta)f(\eta)\,d\eta \\
&= 2\lambda\mu_c\mu_x^2\left[\left(f_1+\frac{\kappa}{\phi}\right)hT(2,0,\eta_0)+\left(\frac{\kappa(1-\phi^3)}{\phi^2(\phi^2-1)}-f_1\right)T(3,0,\eta_0)\right. \\
&\qquad \left. -\left(\frac{\kappa}{\phi^2(\phi^2-1)}\right)T(3,\phi h,\eta_0)+\left(f_1+\frac{\kappa\phi}{\phi^2-1}\right)T(3,h,\eta_0)\right]
\end{aligned}
$$

so that the total variance is:

$$
\begin{aligned}
V(h) &= 2\lambda\mu_c\mu_x^2\left[\left(f_1+\frac{\kappa}{\phi}\right)hT(2,0,0)+\left(\frac{\kappa(1-\phi^3)}{\phi^2(\phi^2-1)}-f_1\right)T(3,0,0)\right. \\
&\qquad \left. -\frac{\kappa}{\phi^2(\phi^2-1)}T(3,\phi h,0)+\left(f_1+\frac{\kappa\phi}{\phi^2-1}\right)T(3,h,0)\right] \\
&\qquad \text{for } \alpha>3 \\
V(h) &\approx 2\lambda\mu_c\mu_x^2\left[\frac{\nu^\alpha h^2\eta_0^{\alpha-1}}{2(\alpha-1)\Gamma(\alpha)}\left(\frac{\kappa}{\phi+1}+f_1\right)\right. \\
&\qquad +\left(f_1+\frac{\kappa}{\phi}\right)hT(2,0,\eta_0)+\left(\frac{\kappa(1-\phi^3)}{\phi^2(\phi^2-1)}-f_1\right)T(3,0,\eta_0) \\
&\qquad \left. -\left(\frac{\kappa}{\phi^2(\phi^2-1)}\right)T(3,\phi h,\eta_0)+\left(f_1+\frac{\kappa\phi}{\phi^2-1}\right)T(3,h,\eta_0)\right] \\
&\qquad \text{for } 1<\alpha\leq3 \\
V(h) &= \infty \\
&\qquad \text{for } \alpha\leq1
\end{aligned}
$$

$$(9)$$

In practice, $\eta_0$ should be chosen as small as is computationally possible since it is term $V_1(h)$ that involves the approximation. Figure 2 shows, for some typical parameter values, how sensitive the expressions of $V(h)$ (blue line), as well as $C(1,h)$ and $S(h)$ (grey and orange lines; and see derivations below), are to the choice of $\eta_0$. As can be seen, values start to be much less sensitive to the change of $\eta_0$ as $\eta_0<0.01$. In this paper, $\eta_0=0.001$ is chosen.

As indicated, similar derivations yield the expressions for the covariance of lag-$k$ $C(k,h)$ and the centred third-order moment $S(h)$ of the RBL1. These expressions can be found in Appendix A2.

For the RBL2, $\mu_x$ is now random, and chosen proportional to $\eta$: $\mu_x=\iota\eta$ so that shorter cells will tend to have greater intensity. The model equations for the RBL2 are therefore obtained from those of the OBL by, first, substituting $\iota\eta$ for $\mu_x$ in the expressions for the OBL model moments, and then proceeding as for the RBL1, i.e. integrating these moments multiplied by the density function of the Gamma distribution of $\eta$. For this model, the constraint upon $\alpha$ obtained when carrying out separate integrations of the additive terms for the moments of the rainfall depth is less stringent than for the the RBL1. If we look at terms the integrands comprise, we find that, for the RBL2, they behave in the neighbourhood of 0 as $\eta^{\alpha-n}$ with $n=1$ for the mean, $n\leq2$ for the variance, covariance and the third-order moment. The integrals of such terms converge as long as $\alpha-n>-1$, i.e. $\alpha>n-1$.





Analogously to the RBL1, it therefore seems that $M(h)$ is finite as long as $\alpha > 0$, $V(h)$, $C(k,h)$ and $S(k,h)$ are finite as long as $\alpha > 1$. But this conclusion is only warranted for the mean. For the other statistics, Taylor expansions of the exponential terms in the neighbourhood of $0$ yield approximations for which the integrals are finite for certain values of $\alpha$ for which the individual additive terms are not integrable. The results are shown in Appendix A3.

## 5   Model calibration and the estimation of standard statistics

**5.1   Standard or block estimation?**

In fitting Poisson-cluster models, it is standard to consider parts of the year separately e.g. seasons or generally months, and to estimate parameter sets for each of these parts. These parts define blocks of data in the full time-series of observed rainfall. The question then arises as to how to deal with data presenting this block structure. The approach used in many papers, and certainly that which was implemented in the early papers that used data from the whole year (e.g. Onof and Wheater (1993)), consisted 240  in treating the data between the blocks of interest (e.g. those corresponding to a given calendar month) as missing data. The standard estimators were then used for the moments of orders 1 to 3 and the proportions of wet periods at the time-scales of interest.

    Work on the representation of the uncertainty in the model parameters (Wheater et al., 2006) and on the optimal weights to be used in the generalised method of moments implemented in the fitting (Jesus and Chandler, 2011) involved calculating 245  statistics for each block of data of interest (e.g. each month of a given calendar month). This led to the use of other estimators, which we refer to as block estimators of the rainfall statistics. These are obtained by calculating the standard statistic of interest for each block of interest and averaging over the blocks (e.g. each month of a given calendar month).

    There is no difference between these two methods as far as the estimate $\hat{M}_{m,h}$ of the mean rainfall intensity for calendar month $m$ and time-scale $h$ is concerned:

$$\hat{M}_{m,h} \;=\; \frac{1}{N_y N_{m,h}} \sum_{j=1}^{j=N_y} \sum_{i=1}^{i=N_{m,h}} Y_{i,j,m}^{(h)}$$

where,

-  $N_y$ is the number of years,

-  $N_{m,h}$ the number of time-steps at scale $h$ in a month of calendar month $m$ (for all months except February for which leap years would lead to a more complicated formula),

-  $Y_{i,j,m}^{(h)}$ extends the notation introduced at the start of the paper: it is the rainfall depth in the $i$-th interval of the $j$-th month of calendar month $m$.





This is however no longer the case with the variance $V_{m,h}$ of the rainfall intensity for calendar month $m$ and time-scale $h$ for which the standard and block (biased) estimates are respectively:

$$\hat{V}_{m,h}^{[1]} = \frac{1}{N_y N_{m,h}} \sum_{j=1}^{j=N_y} \sum_{i=1}^{i=N_{m,h}} (Y_{i,j,m}^{(h)} - \hat{M}_{m,h})^2$$

$$\hat{V}_{m,h}^{[2]} = \frac{1}{N_y N_{m,h}} \sum_{j=1}^{j=N_y} \sum_{i=1}^{i=N_{m,h}} (Y_{i,j,m}^{(h)} - \overline{Y}_{j,m}^{(h)})^2$$

where, $\overline{Y}_{j,m}^{(h)} = \frac{\sum_{i=1}^{i=N_{m,h}} Y_{i,j,m}^{(h)}}{N_{m,h}}$ for $j = 1, ..., N_{m,h}$ are the the (sample) mean depths at time-scale $h$ of the $j$-th month of calendar month $m$.

A little algebra shows a result that is also familiar from Analysis of variance (ANOVA), i.e. that the two estimators are related by:

$$\hat{V}_{m,h}^{[1]} = \hat{V}_{m,h}^{[2]} + \hat{Var}(\overline{Y}_{j,m}^{(h)}) \tag{10}$$

where the added term is the (biased) sample variance of the above averages.

With the third-order centred moments, we also have two distinct expressions for their estimators:

$$\hat{T}_{m,h}^{[1]} = \frac{1}{N_y N_{m,h}} \sum_{j=1}^{j=N_y} \sum_{i=1}^{i=N_{m,h}} (Y_{i,j,m}^{(h)} - \hat{M}_{m,h})^3$$

$$\hat{T}_{m,h}^{[2]} = \frac{1}{N_y N_{m,h}} \sum_{j=1}^{j=N_y} \sum_{i=1}^{i=N_{m,h}} (Y_{i,j,m}^{(h)} - \overline{Y}_{j,m}^{(h)})^3$$

which are related by the following equation:

$$\hat{T}_{m,h}^{[1]} = \hat{T}_{m,h}^{[2]} + \hat{T}(\overline{Y}_{j,m}^{(h)}) + \frac{3}{N_y N_{m,h}} \sum_{j=1}^{j=N_y} (\overline{Y}_{j,m}^{(h)} - \hat{M}_{m,h}) \sum_{i=1}^{i=N_{m,h}} (Y_{i,j,m}^{(h)} - \overline{Y}_{j,m}^{(h)})^2 \tag{11}$$

where $\hat{T}$ is the third-order centred moment.

## 5.2 Are the estimators significantly different? A brief analytical and numerical investigation

### 5.2.1 Block estimation of moments

To estimate the differences between estimators, we can first look at simple examples of independent realisations in which we sample a number of zeroes that corresponds to what is realistic for the proportion dry $p$ at the scale of interest and a simple distribution for the rainfall depths of non-zero rainfalls is assumed, e.g. a Gamma or Generalised Pareto (hereafter GP) (see Menabde and Sivapalan (2000), Montfort and V.Witter (1986)), assuming $N_y = 50$ and $N_{m,h} = 30 \times 24$ (for hourly data).

We found the differences to be less than 1% in the case of either the variances or third-order moments as the additive terms in the equations relating them were found to be very small, for all the relevant time-scales of interest (5 mins to 24 hrs). In the case of the variance, this can be seen by noting that, $\overline{Y}_{j,m}^{(h)}$ has a population variance that is that of the rainfall depths divided





by $N_{m,h}$. So the sample variance $\overline{Y}_{j,m}^{(h)}$ is of the same order of magnitude as $\frac{1}{N_{m,h}}\hat{V}_{m,h}^{[1]}$ which means that the added term in Eq. (10) will be very small. Similar considerations apply to the added terms in Eq. (11).

We also checked that these two methods provide an unbiased estimation of the population second and third-order centred moments, i.e. $V = Var(X)$ and $M3 = E\big[(X - E(X))^3\big]$. These are easily obtained in terms of the corresponding moments ($V_{>0}$ and $M3_{>0}$) of the distribution of non-zero rainfalls (i.e. of a Gamma or GP distribution) using the following easily derivable relations (where $M$ and $M_{>0}$ are the means of the full and the non-zero only distributions):

$$V = (1-p)\big(V_{>0} + pM_{>0}\big) \tag{12}$$

$$M3 = (1-p)\big(M3_{>0} + 3pM_{>0}V_{>0} + p(2p-1)M_{>0}^3\big) \tag{13}$$

### 5.2.2 Block estimation of ratios

However, some authors apply the block estimation approach, not to the moments themselves, but to their ratios, i.e. the coefficient of variation instead of the variance, and the coefficient of skewness instead of the third-order moment (e.g. Kaczmarska et al. (2014)). That means that the block estimator of such ratios is obtained by averaging the estimates of these ratios from the relevant block from each of the years in the data set.

Here, there are no interesting relations to derive between the estimators from the standard and block methods, so we move directly to the simple numerical testing introduced in Sect. 5.2. For $h = 1$, and a proportion of dry periods of $0.9$, we fitted a Gamma and a generalised Pareto (GP) to the non-zero rainfalls at Greenwich (UK). This yielded a Gamma$(1.1629, 0.692)$ and a GPD$(0.1795, 0.654, 0)$ respectively (with the first providing a better fit), with the parameters given, in order as shape, scale and, for the GP, location.

By generating 100 samples of 50 years of hourly data, we find that there is a non-negligible difference between the two estimation methods. Focusing upon the skewnesses, we find $95\%$ simulation bands of $[5.68, 6.65]$ and $[5.50, 6.15]$ for the Gamma samples, i.e. differences that are still small but no longer negligible (of the order of $4\%$).The block estimates clearly underestimate the population skewness of $6.40$. Further, if we look at rainfall from a summer month, e.g. the month of August, these differences are more marked. For the Gamma distribution (Gamma$(0.848, 1.4)$) the bands are now $[6.48, 7.62]$ and $[6.26, 6.92]$ respectively, so a difference that is twice as large for the upper bounds. Again, the population skewness of $7.05$ is underestimated by the block method.

When using the GP distribution (GPD$(0.1795, 0.654, 0)$), the differences between the two methods and the underestimation are starker. The bands are $[7.94, 13.97]$ and $[7.21, 8.55]$ for the standard and block method respectively. The latter underestimates the population skewness of $10.58$ by quite a margin (these results are for the whole year; for August, the GP fit was poor and the population skewness infinite).

These results now need to be confirmed by looking at the case of a time-series with an appropriate correlation structure. This will enable us to ascertain to what extent introducing correlation impacts the performance of the estimators (which are, of course, theoretically designed for samples of independent realisations).





To do this, we use an RBL2 model calibrated to the same data used for the above sampling, namely Greenwich, UK. The
idea is that this rainfall model provides us with a correlation structure that is close enough to the observed correlation to enable
us to conclude as to how one would expect the block estimates to perform with such a correlation structure. We generate
100 samples of 50 years of hourly data with two sets of parameters, obtained from January (winter) and August (summer),
respectively, and the associated theoretical skewnesses calculated from these two parameter sets are 7.69 and 21.73. The 95%
simulation bands obtained from the sampled hourly time series are $[6.80, 7.65]$ and $[12.67, 14.65]$ using the block method,
and $[7.11, 8.44]$ and $[16.70, 28.40]$ using the standard method. In line with the numerical investigation above, we find that,
for both months, the theoretical skewnesses provided by the model equation are underestimated by the block estimate (the
underestimation is particularly significant during summer month), while no significant deviation is obtained for the standard
estimates.

The results we have obtained are indicative of a problem of underestimation of the skewness with the block estimation
method, which is likely to have a significant impact upon the model's ability to reproduce the statistics of extreme rainfall.

## 6    Results and discussion

### 6.1    Block versus standard estimates

Models RBL1 and RBL2 are fitted using the original equations for these models. Although these equations are not shown in
this paper, they are contained in the new sets of equations given above: for each statistic, the first equation given is that found
in the past papers, with its domain of validity for $\alpha$. We note that, for the RBL2, this is $\alpha > 1$ for all statistics, but we imposed
$\alpha > 2$ for this model, in line with the work carried out by Kaczmarska et al. (2014). By using statistical estimators of the
observed statistics based upon the standard and the block estimates as described in Sect. 5 (i.e. the block method takes averages
of ratios), we define two different fitting methods, the standard (sM) and block (bM) fitting methods respectively.

Below, we consider:

– some standard theoretical statistics obtained when the two models are fitted with both methods and how these compare
  with the estimates derived from the observations using the standard and block methods;

– the extreme rainfall depths produced by simulating time-series of identical length to the observations; because of sam-
  pling variability, 250 simulations are carried out and the median is shown;

– the values of the parameters obtained in fitting these models with these two methods

While the mean rainfall depth (which has identical standard and block estimators) is nearly perfectly reproduced by both
methods and models, Fig. 3 shows the differences in the skewness standard and block estimators (crosses and circles respec-
tively).

Consequently, the models fitted to each also yield significantly different skewnesses. Since we know from the preliminary
investigation in Sect. 5 that the standard estimator is much less biased, this means that the block fitting method significantly





underestimates the skewness of the observations. This is an important conclusion with respect to the validity of previous work which has used the block fitting method.

We also note some interesting features of the two models' performance:

– good fits are obtained for RBL1 and RBL2 with the sM for all but the sub-hourly time-scales;

– at the finest time-scale under consideration (i.e. 5 and 10 mins for Bochum and Uccle respectively), there is a considerable
underestimation of the skewness for bM and sM, in particular by the RBL1 model: this confirms the superiority of the RBL2 for fine time-scales noted by Kaczmarska et al. (2014)

While these results confirm the importance of using the standard estimation of observation statistics, this message is not as clear when we consider the reproduction of the coefficient of variation and autocorrelation lag-1, as Fig. 4 and Fig. 5 show.

Due to space constraints, the examination of the effect of changing between bM and sM upon the variances at coarser
time-scales will be presented together with the effect of using the new equations in Sect. 6.3.

From Figures 4 and 5, we note that:

– the sub-hourly coefficients of variation estimated with the standard method are poorly reproduced by the sM as compared with the bM;

– the same is true of the sub-hourly and hourly autocorrelations

These results might seem a little surprising, so it is important to spell out exactly what they mean: the models fitted to the block estimates provide in some cases a better reproduction of the statistics than the models fitted to the standard estimates. This at the very least suggests that the improved reproduction of the skewness by the sM comes at the cost of other statistics being less well reproduced.

The benefit of an improved reproduction of the skewness upon the models' ability to reproduce the frequency of rainfall
extremes at a range of scales is clear, as Fig. 6 shows.

Here, we observe

– sM significantly improves the reproduction of the extremes;

– RBL2 is superior to RBL1, in terms of reproducing the largest extremes in particular at the sub-hourly scales, but also, for instance at the daily scale

The importance of the reproduction of extreme values for the typical applications of such rainfall models means that even taking into account the problems with mean, coefficient of variation and autocorrelation, the sM is preferred. But this leaves us with an important question: are the shortcomings of the sM in reproducing some of these other statistics down to the model or the way it is fitted?

A clue to addressing this question can be obtained by looking at the parameters obtained when fitting with the sM method.
Focusing for instance upon the RBL2, and recalling the constraint $\alpha > 2$, Table 1 shows that the model calibration has yielded values of $\alpha$ on the boundary (as in Kaczmarska et al. (2014)).





Recalling that $\alpha$ is the shape parameter of the distribution of $\eta$, a smaller $\alpha$ leads to a more skewed distribution for this parameter, and thereby also for those which scale with it, such as the storm mean cell intensity in the case of the RBL2. This enables the RBL2 to generate some much more intense cells and thereby yields higher values of the skewness of rainfall depths as we saw above, thereby explaining its superiority over the RBL1.

For model RBL1, the constraint upon $\alpha$ is defined by the limit of validity of the expression for the skewness (e.g. Onof et al. (2013)) i.e. $\alpha > 4$. The parameters that are obtained (Table 2) similarly show that the optimisation algorithm finds the optimum to be near this boundary for most month of the year.

For both models, the fact that the lower limit of parameter $\alpha$ is selected as optimal suggests that a re-examination of the domain of feasibility of the non-linear optimisation carried out when fitting the models is required. This is exactly what the use of the new equations allows us to do as we shall see below. Note that all the above results are confirmed by the Uccle data (see Sect. S1 of the Supplement).

## 6.2 New versus old equations

We now consider the performance of models RBL1 and RBL2 fitted using the new equations for these models presented in this paper. The impact of the use of these equations, if there is any, will be that of an extension of the domain of feasibility of parameter $\alpha$. Since the results presented above have concluded to the superiority of the standard estimates of observation statistics, we shall use this method in what follows. As in the previous section, we examine (i) model parameters, (ii) the reproduction of standard statistics and (iii) the reproduction of extreme rainfall depth statistics.

Here it is useful to start with the parameters shown in Tables 3 and 4. We see that, for most months, in the case of RBL1, and all but one month, in the case of RBL2, the optimal value of $\alpha$ was found outside the domain of feasibility imposed by the equations used in previous research, i.e. $\alpha > 4$ for RBL1 and $\alpha > 1$ for RBL2. For RBL1, we can check that for the months where the new values of $\alpha$ remain inside the old domain of feasibility, the optimal values of $\alpha$ are very similar to those in Table 2. That they are not identical is down to the randomness in the numerical tool used to optimise the objective function.

Looking now at the standard statistics, Figures 7-10 illustrate the impact of relaxing the constraint upon $\alpha$ in terms of the reproduction of the mean, coefficient of variation, autocorrelation lag-1 and skewness of the rainfall depths.

In these figures, aside from the theoretical estimates of the statistics, we show box plots of their sample estimates based upon 250 simulations of 5 minute (and 10 minute for Uccle, see Sect. S2 in Supplement) time-series of length equal to that of the observations (69 and 105 years for Bochum and Uccle, respectively). This is for two reasons. First it is important to check that the equations derived above are correct, which we can do by comparing estimates from these simulations with the theoretical values. Second, by including information about the simulation bands, we show the sampling variability which is useful to judge by how much a model statistic over- or under-estimates the corresponding observation statistic.

What the figures show very clearly is a general improvement of the reproduction of all these statistics, through the use of new equations. The broadening of parameter space thus enables the model to overcome the problem flagged earlier, namely that the attempt to reproduce fine-scale skewnesses led to a deterioration in the reproduction of the other depth statistics.





In particular, we want to draw the reader's attention to the RBL2's ability to reproduce the skewness at all scales of interest. This bodes well for its extreme-value performance which is shown in Fig. 11.

The improvement brought about by the broader parameter space is particular clear at the finest scale of interest (i.e. 5 and 10 minutes for Bochum and Uccle respectively). But we also note an improved reproduction of the extremes of lower return periods for sub-hourly and the hourly time-scale. These are however rather overestimated by both versions of the RBL2 model

for coarser time-scales (i.e. 1 day).

Without looking into the detail of the RBL1 model, the question of its performance as compared to the RBL2, with the new sets of equations in both cases, is illustrated in Fig. 12.

While noting that the above findings are broadly confirmed by the analysis of the Uccle data (see Sect. S2 in Supplement), we can conclude that RBL2 outperforms RBL1 for sub-hourly and hourly time-scales (the 20-min results at Uccle excepted).

Aside from a somewhat better reproduction of low return period extremes by the RBL1 at the 6-hourly scale for Bochum, and since both models provide an equivalent satisfactory reproduction of the daily extreme rainfall depths (RBL2 is better for Uccle), RBL2 is therefore overall to be preferred for the reproduction of observed extremes.

### 6.3    Reproduction of coarse-scale variances

We briefly look at the impacts of the change of estimator of observational statistics and the use of the new equations upon the

reproduction of coarse-scale variability.

Figure 13 shows that:

- as expected, the sM parameter estimates clearly outperform the bM estimates;

- unlike at finer time-scales, there is no clear improvement of the reproduction of the variance for daily-plus scales using new equations;

- beyond 7 days, many, and particularly the largest, of the variances are underestimated in line with the observations made by Marani (2003). This is even clearer in the case of the Uccle data (see Fig. S10 in Supplement)

This suggests that the issue of large-scale variability is probably best addressed by combining Poisson-cluster models with a coarse-scale model that constrain them so that large-scale variances are reproduced.

### 7    Conclusions

This paper has both corroborated certain observations made in previous studies and identified two important issues about how randomised Bartlett-Lewis models are fitted. In summary, first, the importance of the inclusion of the coefficient of skewness among the fitting properties (Cowpertwait, 1998) has indirectly been confirmed: it plays a key role in enabling a good reproduction of rainfall extremes. Second, the new randomised model (RBL2) introduced by Kaczmarska et al. (2014) has an overall better performance than the earlier version originally presented by Rodriguez-Iturbe et al. (1988), in particular





in terms of its ability to reproduce extreme values. Third, we have shown that, while the weights used in the objective function require that estimates of the statistical properties used in the fitting be derived for each single month of the data set (to obtain their variance), in particular in the case of ratios such as the coefficients of variation or of skewness, these estimates should not be used to derive the overall estimates of the relevant statistical property. Rather, the estimates of rainfall statistics for each calendar month are best derived by pooling together all data from the relevant calendar month (with due attention to the

separation between years in the case of the autocovariance) and using the standard sample statistics. Fourth, we have shown that the parameter spaces assumed in previous studies could be extended by relaxing the constraints imposed upon a parameter common to both randomised models ($\alpha$). This improves in particular the RBL2 model's performance in reproducing both standard and extreme value statistics at sub-hourly and hourly time-scales. Fifth, the reproduction of coarse-scale variances (of a few days and more) is improved by using the standard method of estimating observation statistics, but the broader parameter

space does not add much. As a result, we find that these Bartlett-Lewis models still tend rather to underestimate the variability at scales coarser than a week, which provides a confirmation of the wisdom of developing combinations of Bartlett-Lewis models with simple coarse-scale models to capture long-term variability (e.g. see Park et al. (2019) and forthcoming work).

**Appendix A: Formulae for Fitting Properties**

The complete formulae are given here for the selected statistical moments based upon different parameter ranges. These include

mean, variance, lag-$k$ auto-covariance and the third central moment of the discrete time aggregated process of the OBL, RBL1 and RBL2 models.

The definitions of the model parameters used are given below. When a parameter is only valid in some of the models, the models are indicated in square brackets:

– $h$: timescale

– $\lambda$: storm arrival rate

– $\eta$: cell duration parameter [OBL]

– $\alpha$: shape parameter for the Gamma distribution of the cell duration parameter ($\eta$) [RBL1, RBL2]

– $\nu$: scale parameter for the Gamma distribution of $\eta$ [RBL1, RBL2]

– $\beta$: cell arrival rate [OBL]

– $\kappa$: ratio of the cell arrival rate to $\eta$ (i.e. $\beta/\eta$)

– $\gamma$: storm termination rate [OBL]

– $\phi$: ratio of the storm termination rate to $\eta$ (i.e. $\gamma/\eta$)

– $\mu_X = E[X]$: mean cell intensity [OBL, RBL1]





- $\mu_{X^2} = E[X^2]$: mean of squares of cell intensities [OBL, RBL1]

- $\mu_{X^3} = E[X^3]$: mean of cubes of cell intensities [OBL, RBL1]

- $\iota$: ratio of mean cell intensity to $\eta$ (i.e. $\mu_X/\eta$) [RBL2]

- $f_1 = \mu_{X^2}/\mu_X^2$

- $f_2 = \mu_{X^3}/\mu_X^3$

- $\mu_C = 1 + \kappa/\phi$: mean number of cells per storm

**A1    Bartlett-Lewis Rectangular Pulse Model (OBL)**

**Mean**

$$M(h,\eta) = \frac{\lambda h \mu_x \mu_c}{\eta} \tag{A1}$$

**Variance**

$$
\begin{aligned}
V(h,\eta) &= 2\lambda\mu_c\mu_x^2 \left\{ \left(f_1 + \frac{\kappa}{\phi}\right) \frac{h}{\eta^2} + \left(\frac{\kappa}{\phi^2(\phi^2-1)}\right) \frac{1-e^{-\phi\eta h}}{\eta^3} \right.\\
&\qquad \left. + \left(f_1 + \frac{\kappa\phi}{\phi^2-1}\right) \frac{1-e^{-\eta h}}{\eta^3} \right\}
\end{aligned} \tag{A2}
$$

**Covariance at lag $k \geq 1$**

$$
\begin{aligned}
C(k,h) &= \frac{\lambda\mu_c\mu_x^2}{\eta^3} \left\{ \left(f_1 + \frac{\kappa\phi}{\phi^2-1}\right) \left[e^{-\eta(k-1)h} - 2e^{-\eta kh} + e^{-\eta(k+1)h}\right] \right.\\
&\qquad \left. - \left(\frac{\kappa}{\phi^2(\phi^2-1)}\right) \left[e^{-\eta\phi(k-1)h} - 2e^{-\eta\phi kh} + e^{-\eta\phi(k+1)h}\right] \right\}
\end{aligned}
$$

$$\tag{A3}$$

**Third central moment**

$$S(h,\eta) = E\left[\left(Y_i^{(h)} - E(Y_i^h)\right)^3\right] = \frac{\lambda\mu_c\mu_x^3 \sum_{k=1}^{k=8} P_k(\phi,\kappa,\eta,f_1,f_2)}{(1+2\phi+\phi^2)(\phi^4-2\phi^3-3\phi^2+8\phi-4)\phi^3} \tag{A4}$$





where the quantities $P_k \{\phi, \kappa, \eta, f_1, f_2\}$ are given by the following equations:

$$
\begin{aligned}
P_1(\phi, \kappa, \eta, f_1, f_2) &= 6\eta^{-4}e^{-\eta h}\phi^2\left[\phi\kappa^2(2\phi^4 - 7\phi^2 - 3\phi + 2) + 2\phi f_2(\phi^6 - 6\phi^4 + 9\phi^2 - 4)\right. \\
&\quad \left. + \kappa f_1(4\phi^6 - 22\phi^4 - \phi^3 + 25\phi^2 + 4\phi - 4)\right] \\
P_2(\phi, \kappa, \eta, f_1, f_2) &= 6\eta^{-3}e^{-\eta h}\phi^3 h\left[f_2(\phi^6 - 6\phi^4 + 9\phi^2 - 4) + \phi\kappa f_1(\phi^2 - 1)(\phi^2 - 4)\right] \\
P_3(\phi, \kappa, \eta, f_1, f_2) &= 6\eta^{-4}e^{-\eta\phi h}\kappa\left[f_1(-\phi^5 + \phi^4 + 6\phi^3 - 4\phi^2 - 8\phi)\right. \\
&\quad \left. + \kappa(\phi^5 - 3\phi^4 + 2\phi^3 + 14\phi^2 - 8)\right] \\
P_4(\phi, \kappa, \eta, f_1, f_2) &= 6\eta^{-3}e^{-\eta\phi h}h\kappa^2\left[\phi^3(5 - \phi^2) - 4\phi\right] \\
P_5(\phi, \kappa, \eta, f_1, f_2) &= \eta^{-4}\left[-12\phi^3 f_2(\phi^6 - 6\phi^4 + 9\phi^2 - 4)\right. \\
&\quad + \kappa^2(-9\phi^7 + 39\phi^5 + 18\phi^4 - 12\phi^3 - 84\phi^2 + 48) \\
&\quad \left. - 3\phi\kappa f_1(7\phi^7 - 39\phi^5 - 2\phi^4 + 46\phi^3 + 12\phi^2 - 8\phi - 16)\right] \\
P_6(\phi, \kappa, \eta, f_1, f_2) &= \eta^{-3}\left[(6h\phi^3 f_2 + 12h\phi^2\kappa f_1 + 6h\phi\kappa^2)(\phi^6 - 6\phi^4 + 9\phi^2 - 4)\right] \\
P_7(\phi, \kappa, \eta, f_1, f_2) &= 3\eta^{-4}e^{-2\eta h}\phi^4(1 - \phi^2)\left[\phi\kappa^2 + \kappa f_1(\phi^2 - 4)\right] \\
P_8(\phi, \kappa, \eta, f_1, f_2) &= 6\eta^{-4}e^{-(1+\phi)\eta h}\kappa\phi^2(\phi - 2)(\phi - 1)\left[f_1(\phi + 2) - \phi\kappa\right]
\end{aligned}
$$

## A2 Randomised Bartlett-Lewis Rectangular Pulse Model (RBL1)

**Mean**

$$
M(h) = \frac{\lambda h \mu_x \mu_c \nu}{\alpha - 1} \tag{A5}
$$

**Variance**

$$
\begin{aligned}
V(h) &= 2\lambda\mu_c\mu_x^2\left[\left(f_1 + \frac{\kappa}{\phi}\right)hT(2,0,0) + \left(\frac{\kappa(1 - \phi^3)}{\phi^2(\phi^2 - 1)} - f_1\right)T(3,0,0)\right. \\
&\quad \left. - \frac{\kappa}{\phi^2(\phi^2 - 1)}T(3,\phi h, 0) + \left(f_1 + \frac{\kappa\phi}{\phi^2 - 1}\right)T(3, h, 0)\right] \\
&\qquad \text{for } \alpha > 3 \\
V(h) &\approx 2\lambda\mu_c\mu_x^2\left[\frac{\nu^\alpha h^2 \eta_0^{\alpha-1}}{2(\alpha - 1)\Gamma(\alpha)}\left(\frac{\kappa}{\phi + 1} + f_1\right)\right. \\
&\quad + \left(f_1 + \frac{\kappa}{\phi}\right)hT(2, 0, \eta_0) + \left(\frac{\kappa(1 - \phi^3)}{\phi^2(\phi^2 - 1)} - f_1\right)T(3, 0, \eta_0) \\
&\quad \left. - \left(\frac{\kappa}{\phi^2(\phi^2 - 1)}\right)T(3, \phi h, \eta_0) + \left(f_1 + \frac{\kappa\phi}{\phi^2 - 1}\right)T(3, h, \eta_0)\right] \\
&\qquad \text{for } 1 < \alpha \leq 3 \\
V(h) &= \infty \\
&\qquad \text{for } \alpha \leq 1
\end{aligned}
$$

$$\tag{A6}$$





**Covariance at lag $k \geq 1$**

$$
\begin{aligned}
C(k,h) &= \lambda\mu_c\mu_x^2\left\{\left(f_1+\frac{\kappa\phi}{\phi^2-1}\right)[T(3,(k-1)h,0)-2T(3,kh,0)+T(3,(k+1)h,0)]\right.\\
&\quad\left.-\left(\frac{\kappa}{\phi^2(\phi^2-1)}\right)[T(3,\phi(k-1)h,0)-2T(3,\phi kh,0)+T(3,\phi(k+1)h,0)]\right\}\\
&\qquad\text{for }\alpha>3\\
C(k,h) &\approx \lambda\mu_c\mu_x^2\left\{\frac{\nu^\alpha h^2\eta_0^{\alpha-1}}{(\alpha-1)\Gamma(\alpha)}\left(\frac{\kappa}{\phi+1}+f_1\right)\right.\\
&\quad +\left(f_1+\frac{\kappa\phi}{\phi^2-1}\right)[T(3,(k-1)h,\eta_0)-2T(3,kh,\eta_0)+T(3,(k+1)h,\eta_0)]\\
&\quad\left.-\left(\frac{\kappa}{\phi^2(\phi^2-1)}\right)[T(3,\phi(k-1)h,\eta_0)-2T(3,\phi kh,\eta_0,)+T(3,\phi(k+1)h,\eta_0,)]\right\}\\
&\qquad\text{for }1<\alpha\leq3\\
C(k,h) &= \infty\\
&\qquad\text{for }\alpha\leq1
\end{aligned}
$$

(A7)

**Third central moment**

$$
\begin{aligned}
S(h) &= \frac{\lambda\mu_c\mu_x^3\sum_{k=1}^{k=8}Q_k(\phi,\kappa,f_1,f_2,0)}{(1+2\phi+\phi^2)(\phi^4-2\phi^3-3\phi^2+8\phi-4)\phi^3}\\
&\qquad\text{for }\alpha>4\\
S(h) &\approx \frac{\lambda\mu_c\mu_x^3}{(1+2\phi+\phi^2)(\phi^4-2\phi^3-3\phi^2+8\phi-4)\phi^3}\\
&\quad\left[\frac{\nu^\alpha\eta_0^{\alpha-1}h^3}{\Gamma(\alpha)(\alpha-1)}\left(2\kappa^2(\phi^7-3\phi^6+\phi^5+3\phi^4-2\phi^3)+f_2(\phi^9-6\phi^7+9\phi^5-4\phi^3)\right.\right.\\
&\quad\left.+3\kappa f_1(\phi^8-\phi^7-5\phi^6+5\phi^5+4\phi^4-4\phi^3)\right)\\
&\quad\left.+\sum_{k=1}^{k=8}Q_k(\phi,\kappa,f_1,f_2,\eta_0)\right]\\
&\qquad\text{for }1<\alpha\leq4\\
S(h) &= \infty\\
&\qquad\text{for }\alpha\leq1
\end{aligned}
$$

(A8)





and the quantities $Q_k\{\phi, \kappa, f_1, f_2, l\}$ are given by the following equations:

$$
\begin{aligned}
Q_1(\phi, \kappa, f_1, f_2, l) &= 6T(4, h, l)\phi^2 \left[ \phi\kappa^2(2\phi^4 - 7\phi^2 - 3\phi + 2) + 2\phi f_2(\phi^6 - 6\phi^4 + 9\phi^2 - 4) \right. \\
&\quad \left. + \kappa f_1(4\phi^6 - 22\phi^4 - \phi^3 + 25\phi^2 + 4\phi - 4) \right] \\
Q_2(\phi, \kappa, f_1, f_2, l) &= 6T(3, h, l)\phi^3 h \left[ f_2(\phi^6 - 6\phi^4 + 9\phi^2 - 4) + \phi\kappa f_1(\phi^2 - 1)(\phi^2 - 4) \right] \\
Q_3(\phi, \kappa, f_1, f_2, l) &= 6T(4, \phi h, l)\kappa \left[ f_1(-\phi^5 + \phi^4 + 6\phi^3 - 4\phi^2 - 8\phi) \right. \\
&\quad \left. + \kappa(\phi^5 - 3\phi^4 + 2\phi^3 + 14\phi^2 - 8) \right] \\
Q_4(\phi, \kappa, f_1, f_2, l) &= 6T(3, \phi h, l)\kappa^2 h \left[ \phi^3(5 - \phi^2) - 4\phi \right] \\
Q_5(\phi, \kappa, f_1, f_2, l) &= T(4, 0, l) \left[ -12\phi^3 f_2(\phi^6 - 6\phi^4 + 9\phi^2 - 4) \right. \\
&\quad + \kappa^2(-9\phi^7 + 39\phi^5 + 18\phi^4 - 12\phi^3 - 84\phi^2 + 48) \\
&\quad \left. - 3\phi\kappa f_1(7\phi^7 - 39\phi^5 - 2\phi^4 + 46\phi^3 + 12\phi^2 - 8\phi - 16) \right] \\
Q_6(\phi, \kappa, f_1, f_2, l) &= T(3, 0, l) \left[ (6h\phi^3 f_2 + 12h\phi^2 \kappa f_1 + 6h\phi\kappa^2)(\phi^6 - 6\phi^4 + 9\phi^2 - 4) \right] \\
Q_7(\phi, \kappa, f_1, f_2, l) &= 3T(4, 2h, l)\phi^4(1 - \phi^2) \left[ \phi\kappa^2 + \kappa f_1(\phi^2 - 4) \right] \\
Q_8(\phi, \kappa, f_1, f_2, l) &= 6T(4, (1+\phi)h, l)\kappa\phi^2(\phi - 2)(\phi - 1) \left[ f_1(\phi + 2) - \phi\kappa \right]
\end{aligned}
$$

### A3  Randomised Parameter Bartlett-Lewis Rectangular Pulse Model with Dependent Intensity-Duration (RBL2)

**Mean**

$$
M(h) = \lambda h \iota \mu_c \tag{A9}
$$

**Variance**

$$
\begin{aligned}
V(h) &= 2\lambda\mu_c\iota^2 \left[ \left( f_1 + \frac{\kappa}{\phi} \right) h + \left( \frac{\kappa(1 - \phi^3)}{\phi^2(\phi^2 - 1)} - f_1 \right) T(1, 0, 0) \right. \\
&\quad \left. - \left( \frac{\kappa}{\phi^2(\phi^2 - 1)} \right) T(1, \phi h, 0) + \left( f_1 + \frac{\kappa\phi}{\phi^2 - 1} \right) T(1, h, 0) \right] \\
&\quad \text{for } \alpha > 1 \\
V(h) &\approx 2\lambda\mu_c\iota^2 \left[ \frac{\eta_0^{\alpha+1} h^2 \nu^\alpha}{2(\alpha+1)\Gamma(\alpha)} \left( \frac{\kappa}{\phi+1} + f_1 \right) \right. \\
&\quad + \left( f_1 + \frac{\kappa}{\phi} \right) hT(0, 0, \eta_0) + \left( \frac{\kappa(1 - \phi^3)}{\phi^2(\phi^2 - 1)} - f_1 \right) T(1, 0, \eta_0) \\
&\quad \left. - \left( \frac{\kappa}{\phi^2(\phi^2 - 1)} \right) T(1, \phi h, \eta_0) + \left( f_1 + \frac{\kappa\phi}{\phi^2 - 1} \right) T(1, h, \eta_0) \right] \\
&\quad \text{for } -1 < \alpha \le 1
\end{aligned}
\tag{A10}
$$





**Covariance at lag $k \geq 1$**

$$
\begin{aligned}
C(k,h) &= \lambda\mu_c\iota^2 \left\{ \left( f_1 + \frac{\kappa\phi}{\phi^2-1} \right) \left[ T(1,(k-1)h,0) - 2T(1,kh,0) + T(1,(k+1)h,0) \right] \right. \\
&\quad \left. - \left( \frac{\kappa}{\phi^2(\phi^2-1)} \right) \left[ T(1,\phi(k-1)h,0) - 2T(1,\phi kh,0) + T(1,\phi(k+1)h,0) \right] \right\} \\
&\quad \text{for } \alpha > 1 \\
C(k,h) &\approx \lambda\mu_c\iota^2 \left\{ \frac{\eta_0^{\alpha+1}h^2\nu^\alpha}{\Gamma(\alpha)(\alpha+1)} \left( f_1 + \frac{\kappa}{\phi+1} \right) \right. \\
&\quad + \left( f_1 + \frac{\kappa\phi}{\phi^2-1} \right) \left[ T(1,(k-1)h,\eta_0) - 2T(1,kh,\eta_0) + T(1,(k+1)h,\eta_0) \right] \\
&\quad \left. - \left( \frac{\kappa}{\phi^2(\phi^2-1)} \right) \left[ T(1,\phi(k-1)h,\eta_0) - 2T(1,\phi kh,\eta_0) + T(1,\phi(k+1)h,\eta_0) \right] \right\} \\
&\quad \text{for } -1 < \alpha \leq 1
\end{aligned}
$$

(A11)

**Third central moment**

$$
\begin{aligned}
S(h) &= \frac{\lambda\mu_c\iota^3 \sum_{k=1}^{k=8} P_k(\phi,\kappa,f_1,f_2,0)}{(1+2\phi+\phi^2)(\phi^4-2\phi^3-3\phi^2+8\phi-4)\phi^3} \\
&\quad \text{for } \alpha > 1 \\
S(h) &\approx \frac{\lambda\mu_c\iota^3}{(1+2\phi+\phi^2)(\phi^4-2\phi^3-3\phi^2+8\phi-4)\phi^3} \\
&\quad \left[ \frac{\nu^\alpha\eta_0^{\alpha+2}h^3}{\Gamma(\alpha)(\alpha+2)} \left( 2\kappa^2(\phi^7-3\phi^6+\phi^5+3\phi^4-2\phi^3) + f_2(\phi^9-6\phi^7+9\phi^5-4\phi^3) \right. \right. \\
&\quad \left. + 3\kappa f_1(\phi^8-\phi^7-5\phi^6+5\phi^5+4\phi^4-4\phi^3) \right) \\
&\quad \left. + \sum_{k=1}^{k=8} P_k(\phi,\kappa,f_1,f_2,\eta_0) \right] \\
&\quad \text{for } -2 < \alpha \leq 1
\end{aligned}
$$

(A12)





with:

$$
\begin{aligned}
P_1(\phi,\kappa,f_1,f_2,l) &= 6T(1,h,l)\phi^2\left[\phi\kappa^2(2\phi^4-7\phi^2-3\phi+2)+2\phi f_2(\phi^6-6\phi^4+9\phi^2-4)\right. \\
&\quad \left.+\kappa f_1(4\phi^6-22\phi^4-\phi^3+25\phi^2+4\phi-4)\right] \\
P_2(\phi,\kappa,f_1,f_2,l) &= 6T(0,h,l)\phi^3 h\left[f_2(\phi^6-6\phi^4+9\phi^2-4)+\phi\kappa f_1(\phi^2-1)(\phi^2-4)\right] \\
P_3(\phi,\kappa,f_1,f_2,l) &= 6T(1,\phi h,l)\kappa\left[f_1(-\phi^5+\phi^4+6\phi^3-4\phi^2-8\phi)\right. \\
&\quad \left.+\kappa(\phi^5-3\phi^4+2\phi^3+14\phi^2-8)\right] \\
P_4(\phi,\kappa,f_1,f_2,l) &= 6T(0,\phi h,l)h\kappa^2\left[\phi^3(5-\phi^2)-4\phi\right] \\
P_5(\phi,\kappa,f_1,f_2,l) &= T(1,0,l)\left[-12\phi^3 f_2(\phi^6-6\phi^4+9\phi^2-4)\right. \\
&\quad +\kappa^2(-9\phi^7+39\phi^5+18\phi^4-12\phi^3-84\phi^2+48) \\
&\quad \left.-3\phi\kappa f_1(7\phi^7-39\phi^5-2\phi^4+46\phi^3+12\phi^2-8\phi-16)\right] \\
P_6(\phi,\kappa,f_1,f_2,l) &= T(0,0,l)\left[(6h\phi^3 f_2+12h\phi^2\kappa f_1+6h\phi\kappa^2)(\phi^6-6\phi^4+9\phi^2-4)\right] \\
P_7(\phi,\kappa,f_1,f_2,l) &= 3T(1,2h,l)\phi^4(1-\phi^2)\iota^3\left[\phi\kappa^2+\kappa f_1(\phi^2-4)\right] \\
P_8(\phi,\kappa,f_1,f_2,l) &= 6T(1,(1+\phi)h,l)\kappa\phi^2(\phi-2)(\phi-1)\iota^3\left[f_1(\phi+2)-\phi\kappa\right]
\end{aligned}
$$

## Appendix B: Relation between cell intensity parameters

In the model equations, parameters $\mu_x$, $f_1$ and $f_2$ for the RBL1 and $\iota$, $f_1$ and $f_2$ for the RBL2 are three unrelated model parameters only if a three-parameter distribution is chosen for the cell intensity. If a two-parameter distribution is chosen, there will effectively be two unrelated parameters, if a one-parameter distribution is chosen, there will only be one.

Starting with the last case first, the standard choice is the exponential distribution:

$$
f_X(x) = ae^{-ax} \text{ for } x>0
$$

for which:

$$
\begin{aligned}
\mu_x &= 1/a \\
f_1 &= 2 \\
f_2 &= 6
\end{aligned}
$$

So, for the exponential distribution, the only free parameter is $\mu_x$ for the RBL1 and $\iota$ for the RBL2.

Next, we can seek to have more flexibility by using the Gamma distribution:

$$
f_X(x) = \frac{x^{a-1}e^{-x/b}}{b^a\Gamma(a)} \text{ for } x>0
$$





for which:

$$\mu_x = ab$$

$$f_1 = \frac{a+1}{a}$$

$$f_2 = \frac{a^2 + 3a + 2}{a^2}$$

So, for the Gamma distribution there would be two free parameters $\mu_x$ or $\iota$, and $f_1$, with $f_2$ obtained as the following function

of $f_1$:

$$f_2 = 2f_1^2 - f_1$$

The Pareto distribution is a thick-tailed distribution that will produce larger extremes:

$$f_X(x) = \frac{ab^a}{x^{a+1}} \text{ for } x \geq b$$

and for this distribution, we have:

$$\mu_x = \frac{ab}{a-1} \quad \text{(if } a > 1\text{)}$$

$$f_1 = \frac{(a-1)^2}{a(a-2)} \quad \text{(if } a > 2 \text{ i.e. } f_1 > 1\text{)}$$

$$f_2 = \frac{(a-1)^3}{a^2(a-3)} \quad \text{(if } a > 3 \text{ i.e. } f_2 > 1\text{)}$$

For the Pareto distribution there would also be two free parameters $\mu_x$ or $\iota$, and $f_1$, with $f_2$ obtained as the following function of $f_1$:

$$f_2 = \frac{f_1^{3/2}}{f_1^{1/2}(3 - 2f_1) - 2(f_1 - 1)^{3/2}}$$

where we have to have $f_1 < 4/3$ to fulfill the condition $a > 3$.

Finally, a mixed distribution could be chosen, e.g. one which is a mixture of Gamma and Pareto, with weight $\omega$ representing the probability of sampling from a Gamma rather than a Pareto. This would be defined by the following pdf:

$$f_X(x) = \omega \frac{x^{a-1}e^{-x/b}}{b^a \Gamma(a)} + (1-\omega)\frac{cd^c}{x^{c+1}} \text{ for } x \geq d$$

for which the moments are just weighted combinations of those of the Gamma and Pareto distributions:

$$\mu_x = \omega ab + (1-\omega)\frac{cd}{c-1} \quad \text{(if } c > 1\text{)}$$

$$f_1 = \frac{\omega(a+1)ab^2 + (1-\omega)\frac{cd^2}{c-2}}{\left(\omega ab + (1-\omega)\frac{cd}{c-1}\right)^2} \quad \text{(if } c > 2\text{)}$$

$$f_2 = \frac{\omega(a^2 + 3a + 2)ab^3 + (1-\omega)\frac{cd^3}{c-3}}{\left(\omega ab + (1-\omega)\frac{cd}{c-1}\right)^3} \quad \text{(if } c > 3\text{)}$$



Here, we would have three free parameters, $\mu_x, f_1$ and $f_2$ and for the purposes of simulation, we would seek parameters
$\omega, a, b, c$ and $d$ for which the three right-hand sides of the above equations would be equal to $\mu_x, f_1$ and $f_2$, for instance by
minimising a sum of squares. This optimisation problem is underdetermined, but it would make sense to choose at least for the
Gamma parameters $a$ and $b$, values close to values obtained when fitting a Gamma distribution as starting values, or indeed to
fix these two parameters to these values.

*Author contributions.* CO and LW conceptulised the research idea and designed the experiment. CO derived new equations. LW validated
the equations and implemented code and performed the simulations. CO prepared the manuscript with contributions from LW.

*Competing interests.* The authors declare that they have no conflict of interest.

*Acknowledgements.* The authors are grateful for the financial support provided by the Era-Net FloodCitiSense project. The authors would
also like to thank Deutsche Montan Technologie and the Emschergenossenschaft/Lippeverband for providing Bochum rainfall data, and the
Royal Meteorological Institute of Belgium for providing Uccle rainfall data.





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



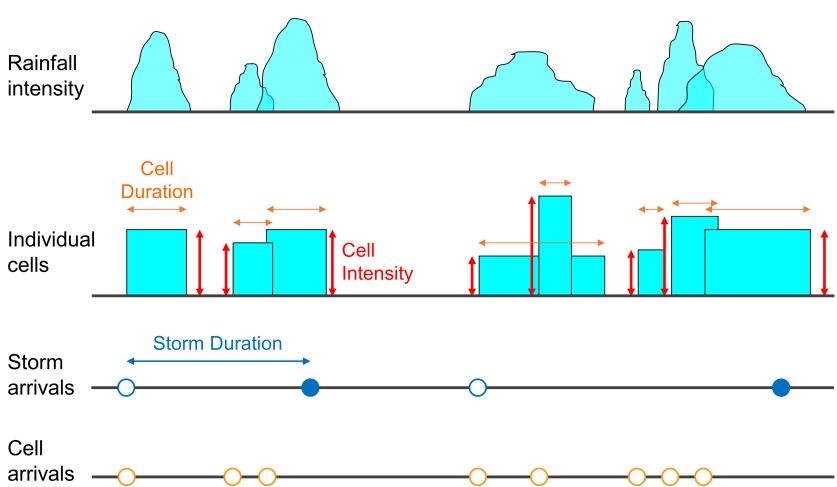

**Figure 1.** Illustration of the conceptualisation of the Bartlett-Lewis Rectangular Pulse model

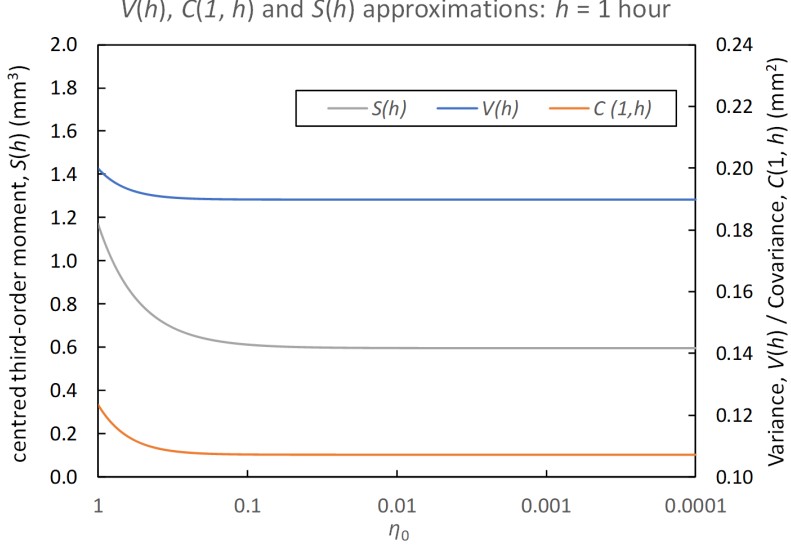

**Figure 2.** Changes of variance ($V(h)$), autocovaraince lag-1 ($C(1,h)$) and the third-order centred moment ($S(h)$) approximations at 1-h timescale ($h = 1$) for $\eta_0 \in (0.0001, 1)$. Parameters used are $\lambda = 0.025$, $\mu_x = 1.3$, $\alpha = 2.5$, $\nu = 0.28$, $\kappa = 0.65$ and $\phi = 0.04$.



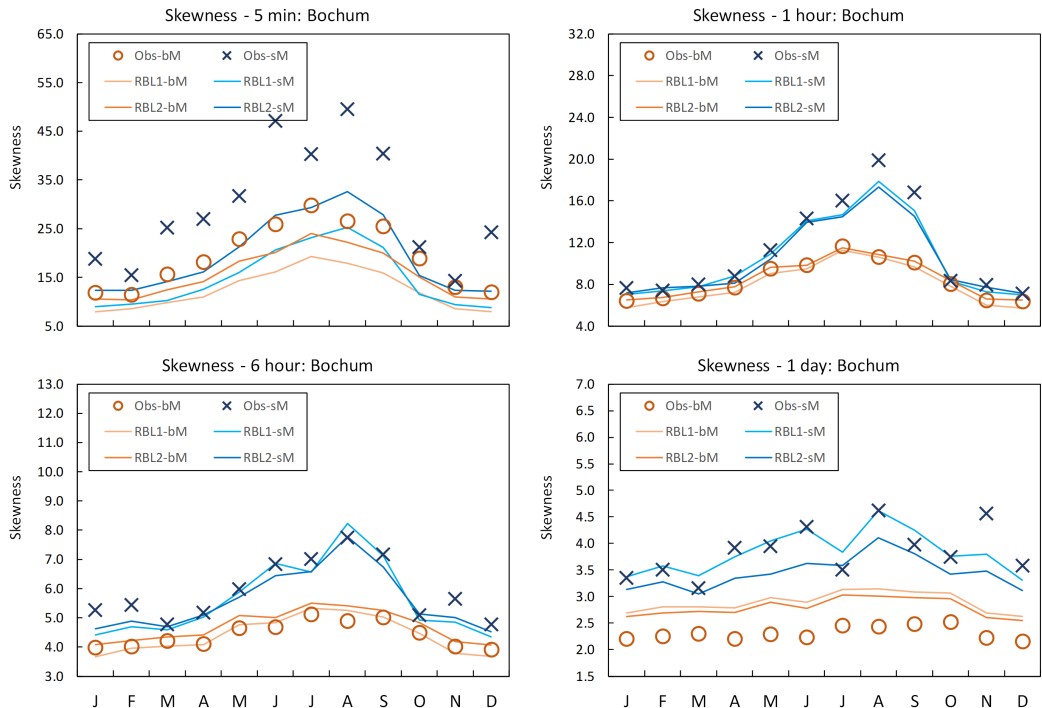

**Figure 3.** Coefficient of skewness by month at Bochum: the observed calculated with block (Obs-bM, orange circle markers) vs. standard (Obs-sM, blue cross markers) methods, the fitted with RBL1 (RBL1-bM, light orange line; RBL1-sM, light blue line) models, and the fitted with RBL2 (RBL2-bM, orange lines; RBL2-sM, blue lines) models



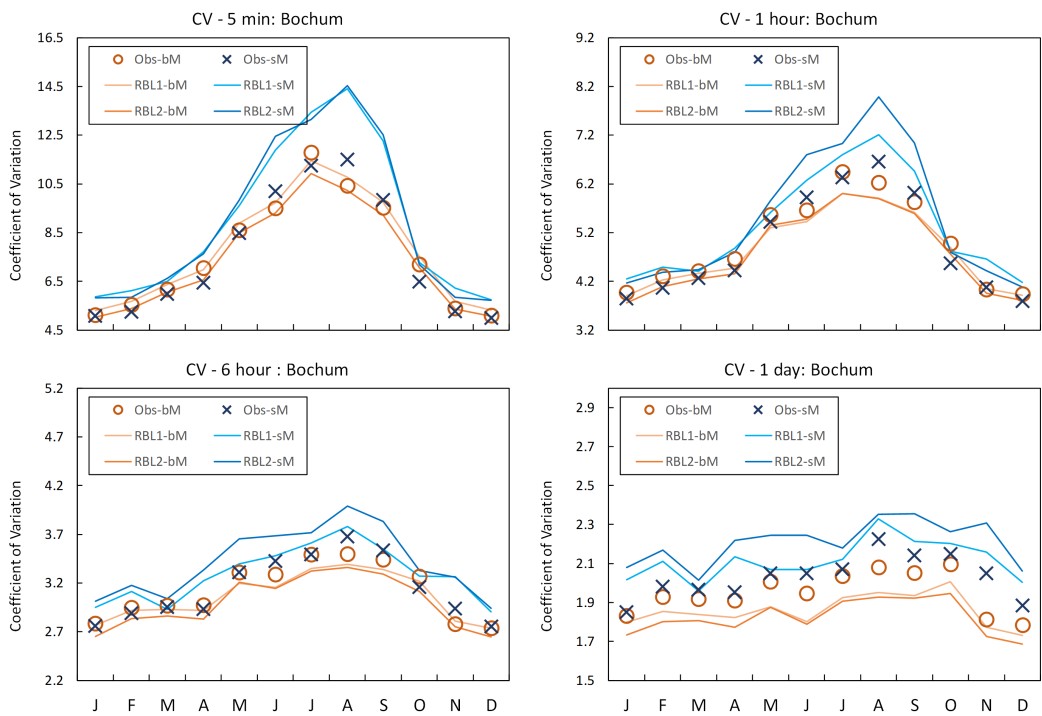

**Figure 4.** Coefficient of variation (CV) by month at Bochum: the observed calculated with block (Obs-bM, orange circle markers) vs. standard (Obs-sM, blue cross markers) methods, the fitted with RBL1 (RBL1-bM, light orange line; RBL1-sM, light blue line) models, and the fitted with RBL2 (RBL2-bM, orange lines; RBL2-sM, blue lines) models

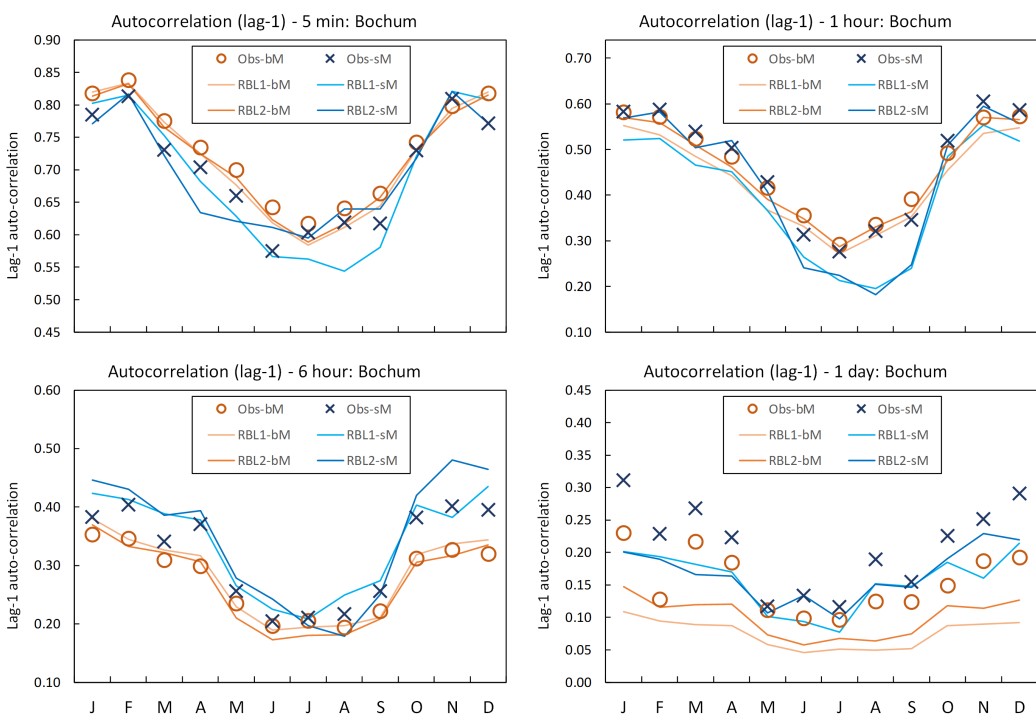

**Figure 5.** Autocorrelation lag-1 by month at Bochum: the observed calculated with block (Obs-bM, orange circle markers) vs. standard (Obs-sM, blue cross markers) methods, the fitted with RBL1 (RBL1-bM, light orange line; RBL1-sM, light blue line) models, and the fitted with RBL2 (RBL2-bM, orange lines; RBL2-sM, blue lines) models

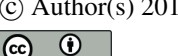

**Figure 6.** Observed (round markers) and simulated (lines) return levels of rainfall at different timescales at Bochum. The simulated is sampled from the RBL1 and RBL2 models fitted with selected statistical properties calculated using bM and sM methods, respectively; and the median return levels obtained from 250 simulations, each of 69 years, are illustrated.





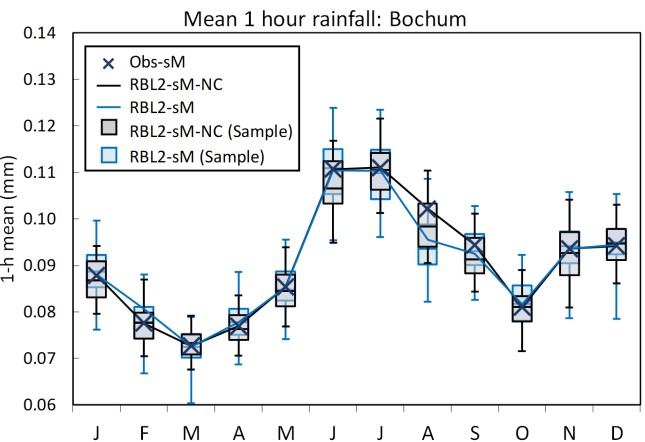

**Figure 7.** Mean 1-hour rainfall depths by month at Bochum: the observed vs. the fitted using RBL2 models with the original and the new solution spaces of $\alpha$ (RBL1-sM, light blue lines and boxplots; RBL2-sM-NC, black lines and boxplots).

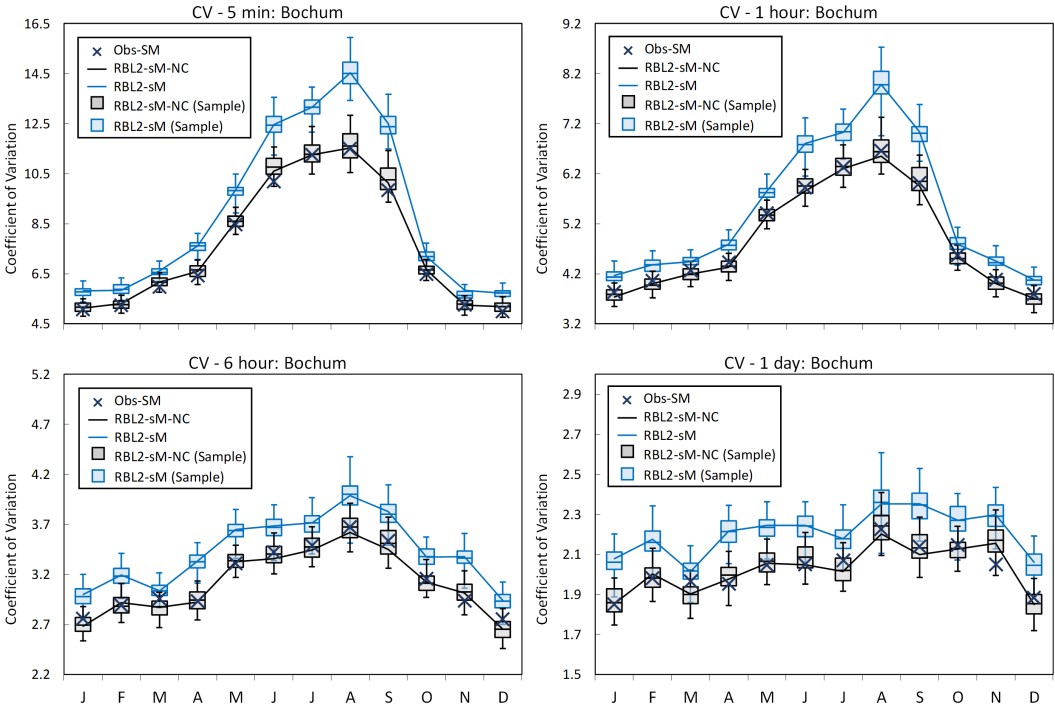

**Figure 8.** Coefficient of variation (CV) by month at Bochum: the observed vs. the fitted using RBL2 models with the original and the new solution spaces of $\alpha$ (RBL2-sM, light blue lines and boxplots; RBL2-sM-NC, black lines and boxplots).

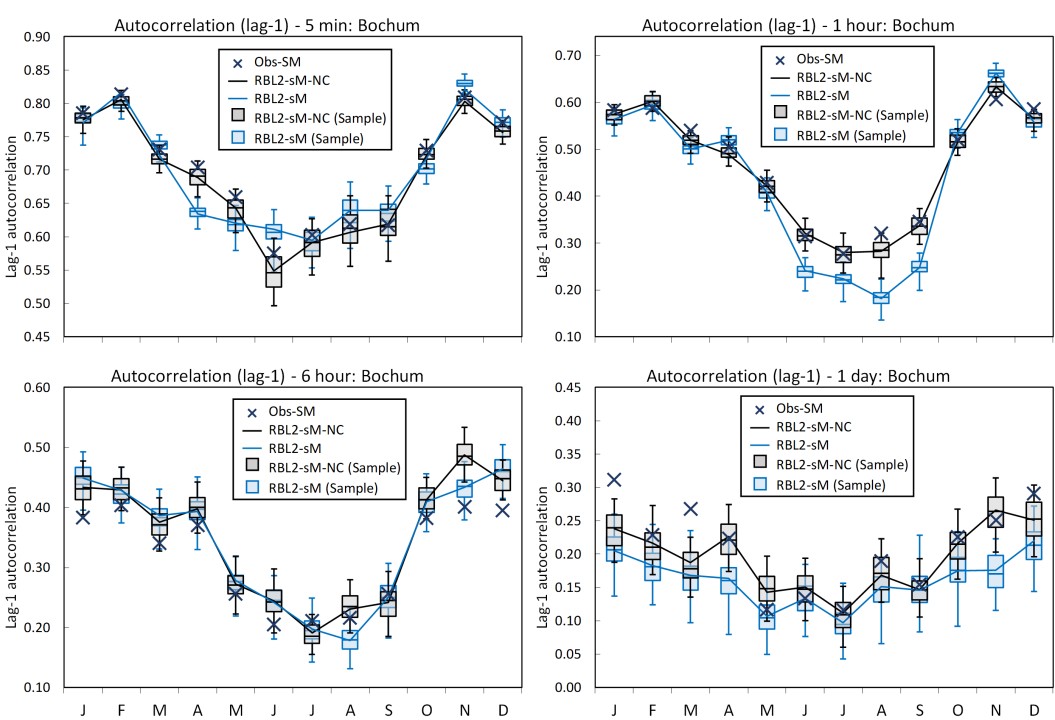

**Figure 9.** Autocorrelation lag-1 by month at Bochum: the observed vs. the fitted using RBL2 models with the original and the new solution spaces of $\alpha$ (RBL2-sM, light blue lines and boxplots; RBL2-sM-NC, black lines and boxplots).



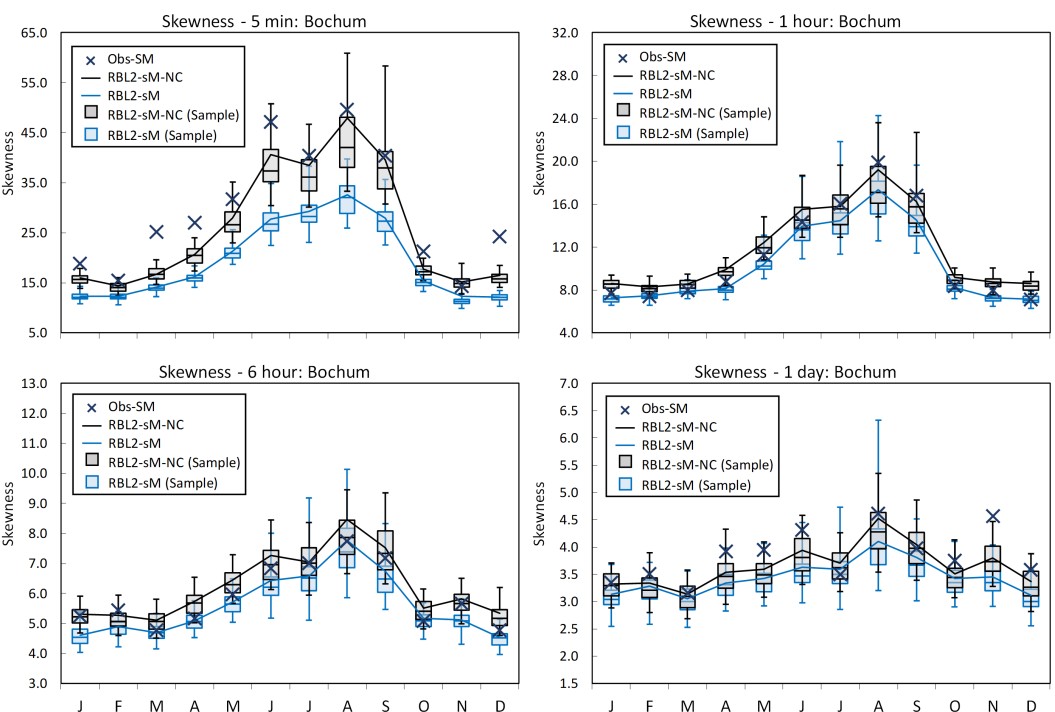

**Figure 10.** Coefficient of skewness by month at Bochum: the observed vs. the fitted using RBL2 models with the original and the new solution spaces of $\alpha$ (RBL2-sM, light blue lines and boxplots; RBL2-sM-NC, black lines and boxplots).

**Figure 11.** Observed (round markers) and simulated (lines) return levels of rainfall at multiple time-scales at Bochum. The simulated is sampled from the RBL2 models fitted with the original (blue lines) and the new (black lines) solution spaces of $\alpha$. The median, 95 and 5 percentile return levels obtained from 250 simulations, each of 69 years, are plotted with solid and dashed lines, respectively.

**Figure 12.** Observed (round markers) and simulated (lines) return levels of rainfall at multiple time-scales at Bochum. The simulated is sampled from the RBL1 (grey lines) and RBL2 (black lines) models fitted with the new solution spaces of $\alpha$. The median, 95 and 5 percentile return levels obtained from 250 simulations, each of 69 years, are plotted with solid and dashed lines, respectively.



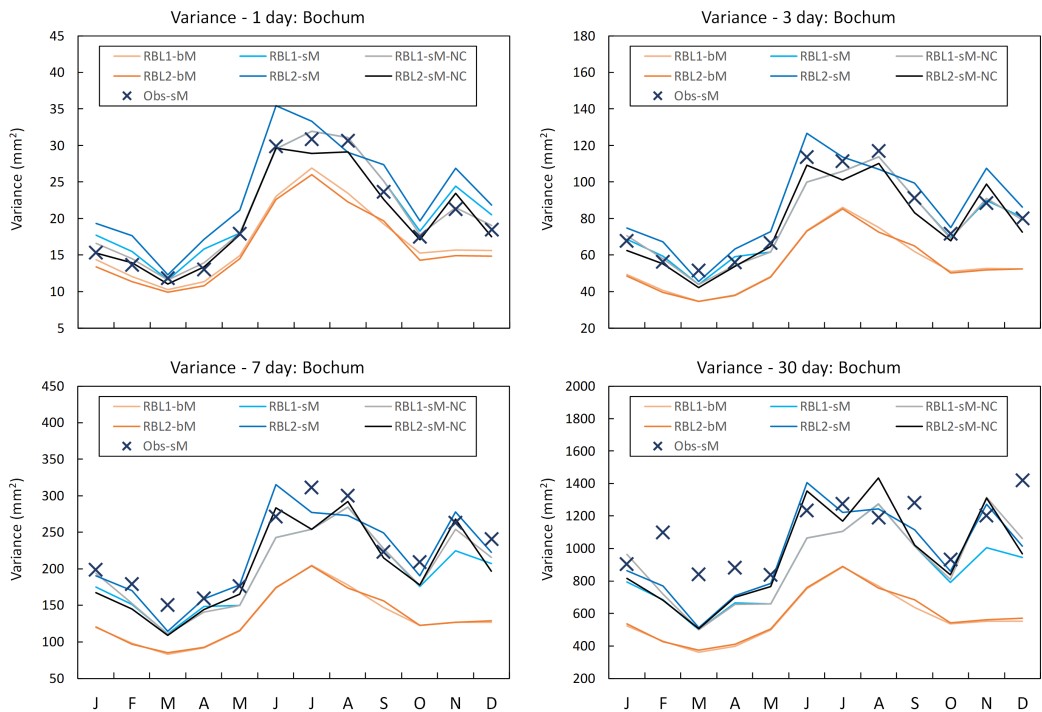

**Figure 13.** Daily Variances by month at Bochum: the observed calculated with standard (Obs-sM, blue cross markers) methods, the fitted with RBL1 (RBL1-bM, light orange line; RBL1-sM, light blue line; RBL1-sM-NC, grey line) models, and the fitted with RBL2 (RBL2-bM, orange lines; RBL2-sM, blue lines; RBL2-sM-NC, black line) models





**Table 1.** Parameters for RBL2-sM model using Bochum gauge data; constraint: $\alpha > 2$

| Month | $\lambda$ | $\iota$ | $\alpha$ | $\alpha/\nu$ | $\kappa$ | $\phi$ |
|-------|-----------|---------|----------|--------------|----------|--------|
| Jan | 0.0131 | 0.2143 | 2.0000 | 5.9436 | 0.7521 | 0.0248 |
| Feb | 0.0124 | 0.2324 | 2.0000 | 4.6243 | 0.9185 | 0.0341 |
| Mar | 0.0152 | 0.2196 | 2.0000 | 6.9889 | 0.5313 | 0.0257 |
| Apr | 0.0125 | 0.2243 | 2.0000 | 11.1401 | 0.4508 | 0.0169 |
| May | 0.0143 | 0.4733 | 2.0000 | 10.2016 | 0.3166 | 0.0273 |
| Jun | 0.0145 | 1.3043 | 2.0000 | 8.3686 | 0.0886 | 0.0182 |
| Jul | 0.0173 | 1.3887 | 2.0000 | 8.9383 | 0.0818 | 0.0228 |
| Aug | 0.0129 | 1.8192 | 2.0000 | 6.9965 | 0.0353 | 0.0115 |
| Sep | 0.0127 | 1.2115 | 2.0000 | 7.3863 | 0.0918 | 0.0183 |
| Oct | 0.0114 | 0.2913 | 2.0000 | 7.1000 | 0.5078 | 0.0216 |
| Nov | 0.0100 | 0.2791 | 2.0000 | 4.4562 | 0.9070 | 0.0279 |
| Dec | 0.0128 | 0.2386 | 2.0019 | 5.6686 | 0.6863 | 0.0230 |

**Table 2.** Parameters for RBL1-sM model using Bochum gauge data; constraint: $\alpha > 4$

| Month | $\lambda$ | $\mu_X$ | $\alpha$ | $\alpha/\nu$ | $\kappa$ | $\phi$ |
|-------|-----------|---------|----------|--------------|----------|--------|
| Jan | 0.0212 | 1.3041 | 4.0000 | 8.7125 | 0.6222 | 0.0318 |
| Feb | 0.0197 | 1.2375 | 4.0000 | 8.0948 | 0.6545 | 0.0357 |
| Mar | 0.0235 | 1.4790 | 4.0270 | 10.8216 | 0.4448 | 0.0279 |
| Apr | 0.0204 | 2.4651 | 4.0000 | 15.4629 | 0.3693 | 0.0218 |
| May | 0.0250 | 4.6258 | 4.3139 | 17.4875 | 0.2567 | 0.0287 |
| Jun | 0.0263 | 10.2162 | 4.4454 | 19.8093 | 0.1195 | 0.0227 |
| Jul | 0.0226 | 13.2244 | 7.5061 | 15.6796 | 0.0843 | 0.0208 |
| Aug | 0.0180 | 14.3142 | 4.5315 | 20.0000 | 0.0558 | 0.0111 |
| Sep | 0.0200 | 9.3123 | 4.4865 | 17.9124 | 0.0879 | 0.0145 |
| Oct | 0.0184 | 2.1201 | 4.0000 | 13.3611 | 0.4589 | 0.0232 |
| Nov | 0.0202 | 1.4692 | 4.0000 | 8.5970 | 0.8587 | 0.0436 |
| Dec | 0.0208 | 1.3561 | 4.0000 | 8.2961 | 0.6026 | 0.0305 |





**Table 3.** Parameters for RBL1-sM-NC model using Bochum gauge data; constraint: $\alpha > 1$

| Month | $\lambda$ | $\mu_X$ | $\alpha$ | $\alpha/\nu$ | $\kappa$ | $\phi$ |
|-------|-----------|---------|----------|--------------|----------|--------|
| Jan | 0.0251 | 1.2511 | 2.7887 | 11.4504 | 0.4977 | 0.0255 |
| Feb | 0.0227 | 1.1887 | 3.1661 | 9.3302 | 0.5924 | 0.0342 |
| Mar | 0.0245 | 1.5670 | 3.7514 | 13.4425 | 0.4390 | 0.0248 |
| Apr | 0.0252 | 2.0977 | 3.1090 | 18.7626 | 0.3714 | 0.0212 |
| May | 0.0250 | 4.6259 | 4.3139 | 17.4875 | 0.2567 | 0.0287 |
| Jun | 0.0265 | 9.7523 | 4.3302 | 20.0000 | 0.1203 | 0.0225 |
| Jul | 0.0226 | 13.2243 | 7.5060 | 15.6796 | 0.0843 | 0.0208 |
| Aug | 0.0185 | 8.3316 | 4.2836 | 20.0000 | 0.0596 | 0.0114 |
| Sep | 0.0200 | 9.3122 | 4.4865 | 17.9123 | 0.0879 | 0.0145 |
| Oct | 0.0197 | 2.0589 | 3.5292 | 14.6293 | 0.4502 | 0.0226 |
| Nov | 0.0274 | 1.3329 | 2.4248 | 12.4683 | 0.6338 | 0.0357 |
| Dec | 0.0273 | 1.3415 | 2.7193 | 15.3848 | 0.5876 | 0.0244 |

**Table 4.** Parameters for RBL2-sM-NC model using Bochum gauge data; constraint: $\alpha > 0$

| Month | $\lambda$ | $\iota$ | $\alpha$ | $\alpha/\nu$ | $\kappa$ | $\phi$ |
|-------|-----------|---------|----------|--------------|----------|--------|
| Jan | 0.0130 | 0.2368 | 0.7408 | 4.1819 | 0.7677 | 0.0280 |
| Feb | 0.0125 | 0.1985 | 0.9747 | 4.2279 | 1.0052 | 0.0333 |
| Mar | 0.0151 | 0.2178 | 0.9812 | 6.1830 | 0.5708 | 0.0271 |
| Apr | 0.0118 | 0.3137 | 0.7190 | 5.6846 | 0.4085 | 0.0206 |
| May | 0.0140 | 0.5276 | 0.6412 | 6.8726 | 0.3718 | 0.0353 |
| Jun | 0.0133 | 1.1797 | 0.4630 | 7.6713 | 0.1305 | 0.0215 |
| Jul | 0.0177 | 1.4427 | 0.6141 | 6.5735 | 0.1063 | 0.0318 |
| Aug | 0.0107 | 1.8582 | 0.4438 | 4.9260 | 0.0664 | 0.0161 |
| Sep | 0.0131 | 1.1473 | 0.4831 | 5.4205 | 0.1612 | 0.0306 |
| Oct | 0.0113 | 0.3041 | 1.0468 | 5.8242 | 0.5131 | 0.0227 |
| Nov | 0.0091 | 0.2344 | 0.8353 | 4.2122 | 1.0394 | 0.0243 |
| Dec | 0.0125 | 0.2575 | 0.7119 | 4.4430 | 0.6700 | 0.0236 |