# Peer review of "Modelling rainfall with a Bartlett-Lewis process: New developments"

_Hydrology and Earth System Sciences, 2019_

## Referee Comment (RC1) · Dongkyun Kim (Referee) · 5 Sep 2019

The focus of this article is to extend the parameter search domain of the randomized Bartlett-Lewis rectangular pulse model. The authors analytically derived the equation representing the first through the third-order moments of the synthetically generated rainfall when the parameter "alpha" is less than one, which, I believe, is a remarkable mathematical endeavor and contribution in our field of rainfall modeling. I have been working on this topic for the last several years, so I came to develop my own version of the model reading the submitted manuscript and had an opportunity to validate it myself using the same Bochum data. Here is what I found:

(1) The authors argue that when the domain of the parameter alpha is extended such that 0<alpah<1, the extreme values can be better represented. To me, as the parameter alpha becomes smaller, the variability of the parameter eta should be also reduced because:

E(eta) = alpha / nu

Var(eta) = alpha^2/nu

, which subsequently reduce the variability of rain cell intensity in the following manner:

Var(miux) = Var(iota * eta) = iota^2 * Var(eta) = iota^2 * alpha^2/nu

I believe that the extreme value should be associated with the tail part of the distribution of miux, but according to the above equation, the tail of the distribution should be thinner.

Therefore, I argue that the reduced value of alpha should improve the model's fitting ability to rainfall characteristics with "more regular" behavior.

(2) The observed annual maxima shown in Figure 11 and Figure 12 seems to be lower than the actual value. According to my calculation, the observed annual maximum of daily rainfall goes upto 90+ mm while the values shown in the figure goes upto only 70mm. I guess this discrepancy came from the way to estimate the annual maxima. In my case, I ran the moving window of a given aggregation interval throughout the 5-minute timeseries over one year to get the maximum value, while the authors aggregated first and then took the maximum.

(3) The parameter estimation process does not seem to have considered rainfall intermittency (e.g. equations for proportion of dry/wet period). If you put the parameter values of Table 4 for the equation of proportion of dry period, the value is almost 0, which means it rains all the time. Please see the figure at the last page of this review.

(4) Please specify the unit of the parameters in the tables. Especially, the parameter iota in the paper confused me because the original Bochum data is in the unit of cm and your iota is in the unit of mm.

It may be also beneficial if you add the column of the objective function values in the tables for the reader's reference.

(4) The parameters with better fit could be estimated. I put the parameter values of Table 4 and validated it myself against the standard statistics, which is shown in the following figure:

[Figure]

I could estimate the better parameter values with the particle swarm optimization algorithm (less underestimation of variance and skewness, and the P0 aligning to 1:1 line). I guess this is because the 6-dimensional objective function has huge multi-modality, so any slope-based optimization method tend to fail to identify the true global minimum.

(5) Regarding the inverse variance weighting scheme (L120-L123), I just have one simple question. Let's say that we consider the proportion of dry period (P0) in the calibration process. The interannual variability of P0 will be very small because it is one minus small value every year (e.g. 0.998, 0.980, 0.950, etc.). Therefore, it will have very high weight. Let's say we consider the proportion of wet period (PW) in the calibration process. The interannual variability will be greater than the first case (e.g. 0.002, 0.020, 0.050, etc.). I think this leads to the controversy because we end up with giving different weight to the same physical property. I am just asking your opinion on this because considering P0 in the calibration process with the inverse-weighting scheme will make the optimization algorithm sacrifice the fit of all the remaining statistics to fit the P0 value.

(6) Regarding the block estimation, the mean of the block values are the estimates of the true statistics, which we can get easily, so I think the parameter estimation should always be performed

based on the true statistics.

---

## Referee Comment (RC2) · Anonymous Referee #2 · 20 Sep 2019

This is a well-written and tightly argued paper that makes an important contribution to the stochastic modelling of precipitation time-series. A principle area of difficulty with the Poisson process models considered is the process of parameter identification. The authors provide a commendably clear review of previous approaches before moving on to point out a hitherto unnoticed flaw in existing methods. They then provide an improved methodology, which is convincingly demonstrated (using some remarkable historical time series data) to significantly improve the ability of the models to reproduce extreme values across a range of temporal scales. This is of importance to a wide range of applications, not least rural and urban flood design. I have just one substantive comment. The paper would benefit greatly from additional material setting out the scope and extent of previous applications for this family of models. The paper is written

for the cognoscenti, but for those not deeply familiar with the models, more context would provide helpful motivation, i.e. lines 14 -19 should be expanded. Other very minor points: Line 54 'the other issue' – would be helpful to remind the reader which of the several issues discussed to this point is meant, to aid readability. L117 I suggest 'the models are generally calibrated. . .' – add 'generally', it's not a universal rule that this method must be used.

---

## Referee Comment (RC3) · Anonymous Referee #3 · 25 Sep 2019

The stated contribution of this paper is twofold: first, to clarify an issue regarding the range of acceptable parameters for a class of stochastic rainfall models that has been in widespread use for some 30 years; and second, to highlight a potential problem with the calibration of these models due to the way that empirical properties are calculated from precipitation data. It is, arguably, quite a specialised topic and the mathematical presentation is quite dense in places, although this on its own should not prevent publication.

Unfortunately however, as far as I can tell the authors' arguments in support of *both* of their main points are flawed. If I understand correctly, their first point can be paraphrased as "there is a potential problem with previous mathematical derivations for these models" (lines 157–158) "so that earlier reported results may be incorrect" (lines

178–181)", but after doing some lengthy mathematics there isn't a problem after all"
(e.g. the expression at line 210). Moreover, I have checked the mathematics for their
"RBL1" model and found that in fact the 'established' derivation is fine: it seems that the
present authors have actually *introduced* the non-problem due to their own approach
to the derivation, which is in itself questionable. I haven't checked the other derivations
in the present paper: however, the apparent errors for the RBL1 model cast doubt on
the credibility of the other results. I also think that the arguments can be simplified
substantially.

The second issue, relating to calculation of empirical properties, again seems to be due
to an error on the authors' part — or, at least, to their use of non-standard procedures
for their empirical calculations. I agree with them that the expressions they're using
should not be used, but I am quite surprised and worried to discover that anybody is
using them at all.

More detailed justification follows.

Mathematical derivations

The authors' concerns, about potential problems with previous mathematical deriva-
tions, are centred around the convergence of integrals that appear in the derivations.
However, the original proponents of the RBL1 model (Rodriguez-Iturbe et al. 1988)
did not use these integrals in their own derivations. In fact, the present authors ap-
pear to have made an error in lines 151–152 where they claim that moments for the
model are obtained by multiplying the corresponding expressions for the OBL model
by the density of $\Gamma(\alpha, 1/\nu)$ and integrating. This would be correct if there was a single
value of $\eta$ for an entire realisation of the process. In the model structure however, each
storm has a different value of $\eta$. In Rodriguez-Iturbe et al. (1988), the derivation of the
variance and covariances does *not* make use of separate integrals as claimed in the
present paper: it just uses the expectation of $\exp(-\eta\phi\tau)/\eta$ where $\tau$ is a temporal lag;

and (correctly) notes that this expectation exists only when $\alpha > 1$. It therefore looks to me as though the apparent problem noted by the present authors may be an artefact of an incorrect — or, perhaps, needlessly complicated — approach to the derivation.

Apart from the error in lines 151–152, the authors' reporting of previous results with "non-valid" estimates for the $\alpha$ parameter (lines 178–181) should have made them stop and think more carefully. The reason is that the model fits are obtained by minimising an expression involving the theoretical model properties. Earlier authors must have calculated the properties for these values of $\alpha$, therefore; but this wouldn't be possible if the integrals diverged (or the algebraic expressions would have produced results that are obviously wrong, such as negative values of $E(X^2)$).

Although I haven't checked *all* of the derivations in the present paper, I suggested above that the arguments could probably be simplified. This is potentially relevant to properties such as the third moment and skewness, which were not presented by Rodriguez-Iturbe et al. (1988) and may indeed require the evaluation of numerous integrals as claimed by the present authors. In view of this, it may be helpful to note that the 'problematic' integrals are all of the form $T(k, u, l)$ in the authors' notation as defined in their equation (5):

$$T(k, u, l) = \frac{\nu^\alpha}{(\nu + u)^\alpha} \frac{\Gamma(\alpha - k, l(\nu + u))}{\Gamma(\alpha)} \, ,$$

where the last term is the ratio of an incomplete to a complete gamma function. The authors' concerns about convergence are all focused on the situation where $l = 0$, because this is where the integrand can become infinite. In this case however, the final numerator in the expression above is a *complete* gamma function so that the expression can be written as

$$T(k, u, l) = \frac{\nu^\alpha}{(\nu + u)^\alpha} \frac{\Gamma(\alpha - k)}{\Gamma(\alpha)} \, .$$

But if $k > 0$ is an integer (which I think it is throughout the paper), we have $\Gamma(\alpha)/\Gamma(\alpha - k) = (\alpha - 1)(\alpha - 2) \ldots (\alpha - k)$ *providing* $\alpha - k$ isn't a negative integer (if it is, then $\Gamma(\alpha - k)$ is undefined). Thus

$$T(k, u, l) = \frac{\nu^{\alpha}}{(\nu + u)^{\alpha}(\alpha - 1) \ldots (\alpha - k)} \; ,$$

which is obviously finite providing none of the terms in the denominator is zero. Unless I've missed something, this seems to resolve the convergence issue much more simply.

Calculation of empirical properties

The authors' second main point relates to the calculation of "block" statistics used for model calibration with uncertainty. They claim that the block estimators of variances and other quantities are biased (e.g. lines 257-260). However, the expressions that they give for these estimators are incorrect because there is no adjustment for degrees of freedom in the denominator in either case: the denominator in the first expression should be $N_y N_{m,h} - 1$ and that in the second expression should be $N_y(N_{m,h} - 1)$. In fact, Section 5.1 of their Jesus and Chandler (2011) reference (cited on line 244) discusses the need for careful treatment of small-sample biases: that discussion would probably be relevant to quantities such as the skewness coefficient, discussed by the present authors at lines 291-294. Jesus & Chandler did not discuss the variance specifically: I assume that this is because the form of an unbiased variance estimator is well-known so they didn't think it needed mentioning. If the variance expressions given by the authors are indeed in standard use, this is worrying: a decent journal is probably not the best place to draw attention to such a basic error, however. The bottom line is that there isn't necessarily a problem with block estimators *per se*; but (as with any other sophisticated technique) if you're going to use them then you need to do it carefully.

---

## Author Comment (AC1) · 20 Nov 2019

We thank Dr Dongkyun Kim for his careful reading of the paper and his perceptive comments.

Here are some responses to the points made by Dr Kim.

**(1) Dr Kim first points out that the model equations suggest that, when parameter $\alpha$ is reduced, the tail of the distribution of cell intensities becomes thinner, which will reduce the estimates of extreme values for given return periods. This in turn should therefore**

*improve the model's fitting ability to rainfall characteristics with "more regular" behavior.*

**Ans:** First, we would like to point out that the variance of $\eta$ is rather: $Var(\eta) = \frac{\alpha}{\nu^2}$. This does not, however impact the point made by the referee, which is an interesting one: if $\alpha$ is smaller, the variance of the distribution of the mean cell intensity in a storm, namely $\mu_X$, is decreased, and this Gamma distribution will therefore have a thinner tail. This might indeed mean that the model is better designed to reproduce the 'regular behaviour' of rainfall.

This would also however seem to imply that it performs less well in terms of extremes, which is not the case. I think that it is probably difficult to draw conclusions insofar as the value of parameter $\nu$ also changes with the value of $\alpha$.

**(2) Dr Kim, who has access to the same data set we used in the paper, points to an apparent discrepancy in the observed annual maxima.**

*The observed annual maxima shown in Figure 11 and Figure 12 seems to be lower than the actual value. According to my calculation, the observed annual maximum of daily rainfall goes upto 90+ mm while the values shown in the figure goes upto only 70mm. I guess this discrepancy came from the way to estimate the annual maxima. In my case, I ran the moving window of a given aggregation interval throughout the 5-minute timeseries over one year to get the maximum value, while the authors aggregated first and then took the maximum*

**Ans:** As the referee indicates, this discrepancy is due to the fact that we have considered daily maxima, while he has been working with 24-hour (moving-window) maxima. We chose not to use moving-window aggregation in order to be consistent with that in Kaczmarska et al. (2014), where the same Bochum dataset was used (see Figure 8 in Kaczmarska et al. (2014)). It is interesting though to note the discrepancy that is obtained, and in principle, we could also include a figure in which we compare the 24-hour extremes from observations and model simulations. Given the fact that the paper is already rather long, and that some additional material will need to be included to address other referee comments, we opt for not including this. In so doing, we are following standard practice.

**(3) Dr Kim raises a very important point about the proportions of dry periods, about which we have not said anything in the paper.**

*The parameter estimation process does not seem to have considered rainfall intermittency (e.g. equations for proportion of dry/wet period). If you put the parameter values of Table 4 for the equation of proportion of dry period, the value is almost 0, which means it rains all the time*

**Ans:** It is indeed correct that the proportions of dry periods (proportions dry) have not been included in the model calibration. That makes them prime candidates for model validation according to the standard practice of stochastic model validation: this involves distinguishing between properties used in the calibration and properties used in validation (rather than splitting the observed data set into a calibration and a validation period).

The proportion of wet periods plots the referee shows in his review indicate some substantial overestimation of the proportions of wet periods, i.e. an underestimation of the proportion dry, by the model over a range of time-scales. This is in fact an issue that we had noted in carrying out model simulations and that can be discussed in both theoretical and sampling aspects.

According to the theoretical form of the proportion dry given in Rodriguez-Iturbe et al (1988) (see equation (2.5)) and its approximation given in Wheater et al. (2006) (see equation (B48) at page 412), we note that the constraint for $\alpha$ is $\alpha > 1$. The constraint for $\alpha$ is however $\alpha > 0$ in the new RBL2-sM-NC model, and, as summarised in Table 4 in the manuscript, the $\alpha$ values we obtained are mostly smaller than or very close to 1. Therefore, the theoretical proportion dry can hardly be derived using the approximate equation given in Wheater et al. (2006).

This issue can however be better addressed through sampling. We had found that the underestimation of the proportion dry is due to the generation of many tiny amounts of rainfall which are not significant for any hydrological application. If we therefore look rather at the proportion of near-dry periods (with rainfall below a small threshold of 0.01 mm per 5-min) the problem disappears at hourly and sub-hourly scales. A comparison is given in Figure 1 of proportion dry statistics derived from 250 simulations of RBL2-sM-NC, RBL2-sM and RBL2-bM models, respectively. As can be seen, the new RBL2-sM-NC can better reproduce proportion dry statistics at 5-min and 1-h timescales than RBL2-sM and RBL2-bM models. However, the RBL2-bM model start to outperform the other two models at multi-hour timescales.

Considering that the paper is already rather long, and we do not use proportion dry for model calibration, we opt for not including this in the main paper. However, we will add this in the supplement.

[Figure]

Figure 1: Proportion dry by month at Bochum: the observed vs. the fitted using RBL2 models with the original and the new solution spaces of $\alpha$ (RBL2-bM, light orange boxplots; RBL2-sM, light blue boxplots; RBL2-sM-NC, black boxplots).

**(4) Dr Kim suggests specifying parameter units and objective function values in the tables.**

*Please specify the unit of the parameters in the tables. Especially, the parameter iota in the paper confused me because the original Bochum data is in the unit of cm and your iota is in the unit of mm. It may be also beneficial if you add the column of the objective function values in the tables for the reader's reference.*

**Ans:** We thank Dr Kim for this suggestion. The units of the parameters will be added to the tables. In addition, a new table (similar to the one below) will be added, summarising the minimum objective function values.

Note that the minimum objective function values of the RBL2-bM model in Kaczmarska et al. (2014) (see Table 2) are given here (in grey font colour). This is to demonstrate that the minimum objective function values obtained in our work are similar to those obtained in the previous research. It is also worth noting that the minimum objective function values of the RBL2-sM-NC model are much lower than those of the RBL2-sM model.

| Model | Jan | Feb | Mar | Apr | May | Jun | Jul | Aug | Sep | Oct | Nov | Dec |
|---|---|---|---|---|---|---|---|---|---|---|---|---|
| RBL1-bM | 85.6 | 66.8 | 89.4 | 93.3 | 127.9 | 105.8 | 107.6 | 126.6 | 114.2 | 92.1 | 102.9 | 83.8 |
| RBL2-bM | 39.5 | 30.1 | 52.1 | 56.2 | 73.0 | 65.2 | 65.6 | 72.8 | 60.4 | 47.0 | 41.0 | 36.6 |
| RBL2-bM* | 39 | 22 | 46 | 63 | 74 | 76 | 92 | 74 | 68 | 47 | 23 | 26 |
| RBL1-sM | 227.5 | 176.7 | 192.1 | 169.1 | 221.9 | 328.5 | 180.3 | 620.3 | 323.9 | 110.1 | 280.4 | 410.0 |
| RBL2-sM | 145.0 | 76.7 | 117.6 | 173.6 | 174.3 | 315.6 | 96.5 | 478.4 | 241.6 | 61.2 | 244.6 | 280.5 |
| RBL1-sM-NC | 186.5 | 169.9 | 192.0 | 149.4 | 221.9 | 328.5 | 180.3 | 620.3 | 323.9 | 107.5 | 104.0 | 348.8 |
| RBL2-sM-NC | 37.4 | 23.7 | 75.7 | 60.9 | 43.7 | 59.1 | 8.2 | 32.9 | 8.5 | 32.4 | 109.2 | 142.6 |

\* The minimum objective function values are obtained from Table 2 in Kaczmarska et al. (2014)

**(5) Dr Kim indicates that using another numerical method, better parameters can be obtained.**

*I could estimate the better parameter values with the particle swarm optimization algorithm (less underestimation of variance and skewness, and the P0 aligning to 1:1 line)*

**Ans:** Again, we thank Dr Kim for his work in reviewing this paper, which amounts to a very thorough and useful investigation. There are two issues here. First, as Dr Kim points out, the objective functions obtained when calibrating these Poisson-cluster rectangular pulse models are highly non-linear and are likely to have many local optima. He is therefore right to point out that non-traditional numerical methods such as Particle Swarm optimisation are likely to be very useful to avoid an iterative algorithm converging to a non-global local optimum. Second, however, the statistics used in the fitting by Dr Kim are different from the ones we used – and in particular, they are likely to include the proportion dry. Aside from the issue of improved reproduction of the proportion dry, we would have to look at whether the other statistics are significantly improved.

Based upon Dr Kim's results, we agree that, in this case, Particle Swarm optimisation is a better solver than our numerical method that, as described in the paper, combines the Simulated Annealing and the downhill simplex Nelder-Mead algorithms. We are keen to try the Particle Swarm optimisation method in our future work. However, we would like to highlight that the main contribution of this paper is the re-investigation of the key parameter constraint for the RBL models and the associated new formulation; and we believe that the impact of this change in preserving sub-hourly extreme statistics is likely to be more significant than that resulting from a better numerical solver.

**(6) Dr Kim raises an interesting point about the fact that there seems to be a difference in the role that the proportion dry and the proportion wet would play in the objective function, when we use the weights that are recommended for these generalised methods of moments.**

*Let's say that we consider the proportion of dry period (P0) in the calibration process. The interannual variability of P0 will be very small because it is one minus small value every year (e.g. 0.998, 0.980, 0.950, etc.). Therefore, it will have very high weight. Let's say we consider the proportion of wet period (PW) in the calibration process. The interannual variability will be greater than the first case (e.g. 0.002, 0.020, 0.050, etc.)*

**Ans:** I think there is a misconception here in thinking that the variability of the proportion dry will be different from that of the proportion wet. Indeed, we always have $Var(1 - X) = Var(X)$ so if $X$ is the proportion dry, then $1 - X$ is the proportion wet, and they both have the same variance, and will therefore be given the same weight in the objective function.

**(7)   Dr Kim makes a point about the block estimates**

*Regarding the block estimation, the mean of the block values are the estimates of the true statistics, which we can get easily, so I think the parameter estimation should always be performed based on the true statistics*

**Ans:** We agree with the last part of the statement if 'true statistics' means 'best available estimates of the population statistics' in some sense of best (probably including non-biased, maybe also with minimal variance). But the block estimation method introduces some bias for instance.

**References**

Kaczmarska, J., Isham, V., and Onof, C.: Point process models for fine-resolution rainfall, Hydrological Sciences Journal, 59, 1972–1991, https://doi.org/10.1080/02626667.2014.925558, 2014.

Rodriguez-Iturbe, I., Cox, D. R., and Isham, V.: A point process model for rainfall: further developments, Proceedings of the Royal720Society of London A: Mathematical, Physical and Engineering Sciences, 417, 283–298, https://doi.org/10.1098/rspa.1988.0061, http://rspa.royalsocietypublishing.org/content/417/1853/283, 1988

Wheater, H. S., Isham, V. S., Chandler, R. E., Onof, C., Bellone, E., Prudhomme, C., and Crooks, S.: Improved methods for national spatial-temporal rainfall and evaporation modelling for BSM, Tech. Rep., DEFRA, randd.defra.gov.uk/Document.aspx?Document=FD2105_6227_PR.pdf, 2006.

---

## Author Comment (AC2) · 20 Nov 2019

We thank Referee #2 for her/his positive comments. We agree that this is indeed quite a remarkable dataset to have access to. We also agree that this work is relevant to a wide range of applications beyond flood design.

**(1) The referee makes an important comment about the context setting in the paper.**

*The paper would benefit greatly from additional material setting out the scope and extent of previous applications for this family of models. The paper is written for the cognoscenti, but for those not deeply familiar with the models, more context would provide helpful motivation, i.e. lines 14 -19 should be expanded.*

**Ans:** We thank the referee for pointing that out. The paper is indeed light on information about the applicability of these models. We therefore add some sentences to provide further context setting to motivate the work and show its importance in relation to applications.

**(2) The two other minor comments made by the referee are addressed by making the required changes in the text.**
- Line 54: This sentence has been changed to: *The other issue **(i.e., the underestimation of rainfall variability at scales equal to or larger than a few days)** had not so far received much attention although it is in fact of clear practical import.*
- Line 117: This sentence has been changed to: *The models are **generally** calibrated....*

---

## Author Comment (AC3) · 11 Dec 2019

We thank Referee #3 for her/his detailed comments on the paper.

Two important sets of issues are raised, and in both cases, we thank the referee for a careful reading of the paper which brings up issues that certainly need to be addressed if the message in the paper is to be conveyed clearly.

A. Derivations of the equations for moments of the rainfall depth.

**A.1. The reviewer argues that no separation of integrals occurred in the original papers and that the condition on $\alpha$ remains valid (i.e. that it must be larger than 1):**

*In the model structure however, each storm has a different value of η. In Rodriguez-Iturbe et al. (1988), the derivation of the variance and covariances does not make use of separate integrals as claimed in the present paper: it just uses the expectation of $exp(-\eta\varphi\tau)/\eta$ where τ is a temporal lag; and (correctly) notes that this expectation exists only when $\alpha > 1$. It therefore looks to me as though the apparent problem noted by the present authors may be an artefact of an incorrect — or, perhaps, needlessly complicated — approach to the derivation.*

**Ans:** In response, we would first like to point out that, if one looks at the first time these equations were derived, in the paper the referee refers to (Rodriguez-Iturbe et al., 1988), the derivation does in fact involve separating the integrals. This can be seen by looking at equation (2.2): the expectation that is calculated is obtained by separating into additive terms and taking the expectation of each of these terms (thus integrating each additive term separately).

This might seem fine, and indeed, one of the present authors rederived these equations in the past without noticing any problem. But in fact the integral of a sum of terms is only equal to the sum of the integrals of each additive term when the latter are finite. When the latter are infinite (which happens for $\alpha \leq 3$ not 1, because the pdf of the Gamma distribution is multiplied by this term as explained in the paper), this is not necessarily the case. That is, it is, in general, possible that the integral of the sum should be finite while the integrals of the additive terms are infinite.

Here is an example to illustrate this: consider the integral:

$$I = \int_0^x \frac{e^{\omega t} - e^{-\sigma t}}{t} \, dt$$

If we consider the sum of the two integrals:

$$I_1 = \int_0^x \frac{e^{\omega t}}{t} \, dt \text{ and } I_2 = \int_0^x \frac{-e^{-\sigma t}}{t} \, dt$$

we have two divergent integrals and we cannot say what this sum is $(+\infty - \infty)$.

But $I \neq I_1 + I_2$ because, using Taylor expansions, we see that $I$ is in fact finite:

$$I = \int_0^x \frac{1 + \omega t - 1 + \sigma t + o(t)}{t} \, dt = \int_0^x \left( \omega + \sigma + o(1) \right) dt$$

So, for $x$ small:

$$I \cong (\omega + \sigma)x$$

This situation is analogous to that in the paper, with $x = \eta_0$. That is, we have shown that the integrand can be approximated by sums of Taylor series which, in the neighbourhood of $\eta = 0$ have terms that cancel out, so that the convergence of the integral as a whole is not defined by the convergence of the expectation of $exp(-\eta\varphi\tau)/\eta$ or $exp(-\eta\tau)/\eta$. As we show in the paper, in the vicinity of $\eta = 0$, the integrand of the variance of the RBL1 model is a term in $\eta^{\alpha-2}$ which converges as long as $\alpha > 1$.

To avoid this misunderstanding, we add some text to explain the key point that one cannot just treat integrals of additive terms separately when they are infinite.

**A.2. The referee then raises a good point about the results found by previous authors and the claim that we make that they could be erroneous.**

*the authors' reporting of previous results with "non-valid" estimates for the α parameter (lines 178–181) should have made them stop and think more carefully. The reason is that the model fits are obtained by minimising an expression involving the theoretical model properties. Earlier authors must have calculated the properties for these values of α, therefore; but this wouldn't be possible if the integrals diverged (or the algebraic expressions would have produced results that are obviously wrong, such as negative values of $E(X^2)$).*

**Ans:** There are two senses in which previous results might have been erroneous. The first, and most important one is that the domain of valid values of $\alpha$ was smaller than it need be in cases where the authors used the restrictions upon $\alpha$ required by the separate integration of the additive terms. This is the main reason for the work in this paper: it will allow a broader domain of values of $\alpha$ to be used.

But second, and this is the case that the referee is alluding to, the issue of convergence of the integrals was often not addressed by the authors (including some published with one of the present authors as co-author). This would however not necessarily be picked up when carrying out the optimisation, because all it means is that the expression for the corresponding moment would not have had a numerical value that was that of the model skewness. This would not have involved any negative numbers (only in some exponents). So, for instance, if we look at the claim we made on lines 173-179, we are drawing attention to the fact that, for the convergence of variance and covariance of the RBL1, the condition $\alpha > 3$ is required. The papers listed have some parameters smaller than 3. How does that translate in terms of the kind of quantities these authors would have found for, say, the variance. Well, if we look at the equation for the variance they would have used (see lines 203-205), it involves some values of the function $T$: $T(2, \dots, \dots)$ and $T(3, \dots, \dots)$. Going back to the definition of this function $T$ (lines 157-158), we see that when $\alpha$ is less than 3, the term in the denominator, i.e. $(v + u)^{\alpha-k}$ has an exponent $\alpha - k$ which is $\alpha - 2$ for $k = 2$ and $\alpha - 3$ for $k = 3$. These exponents are less than 1 when $\alpha \leq 3$. But we will, as the referee points out, get some negative terms because of the negative $\alpha - 3$ but overall, we find positive variances. For instance, looking at the parameters $A_1$ and $A_2$ found by Rodriguez-Iturbe et al. (1988),

$$A_1 = \frac{\lambda \mu_C v^\alpha}{(\alpha - 1)(\alpha - 2)(\alpha - 3)} \left[ E(X^2) + \frac{\kappa \varphi \mu_X^2}{\varphi^2 - 1} \right]$$

$$A_2 = \frac{\lambda \mu_C \kappa \mu_X^2 \nu^\alpha}{\varphi^2(\varphi^2 - 1)(\alpha - 1)(\alpha - 2)(\alpha - 3)}$$

we find that term $A_1$ on p.288 of that paper is negative when $1 < 2 < \alpha < 3$, while $A_2$ is positive. This will in fact generally be the case as can be seen from the following reasoning. Recall that $\varphi$ is the ratio of the mean cell duration in the storm and the mean duration of storm activity. In practice (by which I mean, for a physically realistic set of parameters), we will therefore have $\varphi < 1$. From this we can conclude that $A_2 > 0$ when $1 < 2 < \alpha < 3$.

Next, consider $\kappa$: we will in practice have $\kappa > \varphi$, because $\kappa$ is the ratio of the mean cell duration in the storm and the mean cell interarrival time in the storm, whereby the latter is less than the mean duration of storm activity. With these relations, we can show that the terms in square brackets is positive, and therefore $A_1 < 0$. Starting with the fact that $E(X^2) > \mu_X^2$ since the difference between left- and right-hand side is the variance of the cell depth $X$, we can then write the following for the terms in square brackets:

$$E(X^2) + \frac{\kappa\varphi\mu_X^2}{\varphi^2 - 1} = E(X^2) - \frac{\kappa\varphi}{1 - \varphi^2}\mu_X^2 > E(X^2)\left(1 - \frac{\kappa\varphi}{1 - \varphi^2}\right)$$

since the coefficient of $\mu_X^2$ is negative. Further, we get:

$$E(X^2)\left(1 - \frac{\kappa\varphi}{1 - \varphi^2}\right) = E(X^2)\left(\frac{1 - \varphi^2 - \kappa\varphi}{1 - \varphi^2}\right) > E(X^2)\left(\frac{1 - \varphi^2 - \kappa^2}{1 - \varphi^2}\right) > E(X^2) > 0$$

The expression for the variance is:

$$\text{var}\left(Y_i^{(h)}\right) = 2A_1\{(\alpha - 3)h\nu^{2-\alpha} - \nu^{3-\alpha} + (\nu + h)^{3-\alpha}\}$$
$$- 2A_2\{\varphi(\alpha - 3)h\nu^{2-\alpha} - \nu^{3-\alpha} + (\nu + \varphi h)^{3-\alpha}\}$$

in which we have just shown that $A_1 < 0$ and $A_2 > 0$.

The terms in curly brackets are both negative for the parameters we have looked at. The key to the sign of the variance will therefore be the relative sizes of $|A_1|$, $|A_2|$ and of the associated curly bracket terms (denote $|C_1|$ and $|C_2|$ here). Values for these expressions with a typical parameter set for which $2 < \alpha < 3$ are given in Table 1. As can be seen, with typical parameter values: $|A_1| < |A_2|$ and $|C_1| \gg |C_2|$. Therefore, we still get a positive expression for the variance. This means that this would not be picked up as anomalous in calibrating the model.

To clarify these issues, we introduce a sentence explaining the two types of problems and emphasising the one about non-optimality because of an unnecessarily narrow parameter space (which is currently not sufficiently clear in the paper). Since we cannot estimate the impact of the second problem without looking at the particular data sets used in past papers, we rephrase the comment we make about previous studies so that it indicates that one cannot be sure that the parameters found in these previous studies are optimal.

Table 1: Calculations of the variance expression, given in Rodriguez-Iturbe et al. (1988), and the associated parameters at 1-h timescale ($h = 1$) for $2 < \alpha < 3$. Other parameters used are $\lambda = 0.025$, $\mu_x = 1.3$, $\nu = 0.28$, $\kappa = 0.65$ and $\varphi = 0.04$.

| $\alpha$ | $C_1$ | $A_1$ | $C_1 \times A_1$ | $C_2$ | $A_2$ | $C_2 \times A_2$ | $\mathrm{var}\left(Y_i^{(h)}\right)$ |
|---|---|---|---|---|---|---|---|
| 2.1 | -1.0871 | -1.0031 | 1.0904 | -0.0003 | 206.7770 | -0.0574 | 2.2958 |
| 2.2 | -1.1085 | -0.4554 | 0.5048 | -0.0006 | 93.8752 | -0.0524 | 1.1145 |
| 2.3 | -1.1159 | -0.2820 | 0.3147 | -0.0008 | 58.1307 | -0.0482 | 0.7257 |
| 2.4 | -1.1048 | -0.2017 | 0.2229 | -0.0011 | 41.5858 | -0.0445 | 0.5348 |
| 2.5 | -1.0703 | -0.1592 | 0.1703 | -0.0013 | 32.8071 | -0.0414 | 0.4234 |
| 2.6 | -1.0060 | -0.1368 | 0.1377 | -0.0014 | 28.2088 | -0.0386 | 0.3526 |
| 2.7 | -0.9046 | -0.1296 | 0.1172 | -0.0014 | 26.7155 | -0.0362 | 0.3069 |
| 2.8 | -0.7573 | -0.1415 | 0.1071 | -0.0012 | 29.1578 | -0.0340 | 0.2823 |
| 2.9 | -0.5533 | -0.2098 | 0.1161 | -0.0007 | 43.2381 | -0.0321 | 0.2963 |

**A.3. The referee proposes a simplification of the equations by using the fact that ratios of Gamma functions can lead to simpler expressions.**

*The authors' concerns about convergence are all focused on the situation where $l = 0$, because this is where the integrand can become infinite. In this case however, the final numerator in the expression above is a complete gamma function so that the expression can be written as*

$$\Gamma(k, u, l) = \frac{\nu^\alpha}{(\nu + u)^\alpha} \frac{\Gamma(\alpha - k)}{\Gamma(\alpha)}$$

*But if $k > 0$ is an integer (which I think it is throughout the paper), we have $\Gamma(\alpha)/\Gamma(\alpha - k) = (\alpha - 1)(\alpha - 2) \dots (\alpha - k)$ providing $\alpha - k$ isn't a negative integer (if it is, then $\Gamma(\alpha - k)$ is undefined). Thus*

$$\Gamma(k, u, l) = \frac{\nu^\alpha}{(\nu + u)^\alpha (\alpha - 1) \dots (\alpha - k)}$$

*which is obviously finite providing none of the terms in the denominator is zero. Unless I've missed something, this seems to resolve the convergence issue much more simply.*

**Ans:** This simplification is indeed well known and indeed, going back again to the Rodriguez-Iturbe et al., (1988), it was used for instance in equation (2.4): this is why no Gamma functions appear in the expressions of coefficients $A_1$ and $A_2$, but rather products such as $(\alpha - 1)(\alpha - 2)(\alpha - 3)$.

As the referee notes we can only use the Gamma function (and the simplifications that follow) when the variable in the Gamma function is non-negative (there is actually an extension of the Gamma function to negative non-integer numbers, but there is no obvious justification for using it: the Gamma function is introduced in the calculations because of the Gamma distribution, which requires positive parameters). This means, as stated by the referee that $\Gamma(\alpha - k)$ is defined as long as $\alpha > k$. This is precisely the issue that our paper is dealing with: this condition arises when calculating the integrals separately, and unnecessarily restricts the domain of feasibility for the minimisation of the objective function. By not treating the integrals separately which leads to using these (complete) Gamma functions, we show that the domain of possible values of $\alpha$ is broader.

Since it is likely that some readers will also wonder why we did not use complete Gamma functions and the simplifications which ensue, we add a sentence to explain this.

**B. The block estimators**

**The referee points to the fact that we have not used the standard unbiased estimators of the variance and that, if the problem that we are flagging is that some previous authors have not done this, then it is not worth being discussed extensively, if at all.**

*The authors' second main point relates to the calculation of "block" statistics used for model calibration with uncertainty. They claim that the block estimators of variances and other quantities are biased (e.g. lines 257-260). However, the expressions that they give for these estimators are incorrect because there is no adjustment for degrees of freedom in the denominator in either case: the denominator in the first expression should be $N_y N_{m,h} - 1$ and that in the second expression should be $N_y(N_{m,h} - 1)$. In fact, Section 5.1 of their Jesus and Chandler (2011) reference (cited on line 244) discusses the need for careful treatment of small-sample biases: that discussion would probably be relevant to quantities such as the skewness coefficient, discussed by the present authors at lines 291-294. Jesus & Chandler did not discuss the variance specifically: I assume that this is because the form of an unbiased variance estimator is well-known so they didn't think it needed mentioning. If the variance expressions given by the authors are indeed in standard use, this is worrying: a decent journal is probably not the best place to draw attention to such a basic error, however. The bottom line is that there isn't necessarily a problem with block estimators per se; but (as with any other sophisticated technique) if you're going to use them then you need to do it carefully.*

**Ans:** We fully agree that, indeed, if the issue were the bias that is introduced by using $N_g N_{m,h}$ rather than $N_g N_{m,h} - 1$ and $N_g(N_{m,h} - 1)$, then this would not be worth discussing. As the referee points out, the differences between estimators using the first rather than the second coefficients are of relevance to small samples.

However, we are not concerned with small samples here. The smallest sample that might be involved would be of size 30 (in the case of the daily time-scale, for the block estimation method, and for all other time-scales used in the fitting, they are considerably larger). The estimators which do not use the '-1' adjustments are biased (and we indicate that, at least for one of them, after equation (10)). We would not agree that they are 'wrong': they just have a bias that can be corrected by using the '-1' adjustments pointed out by the referee, but they are asymptotically unbiased. And for samples that are, at worst, of size 30, the bias is not of much practical relevance (e.g. typically a few percent).

Still, one might ask why we did not use the unbiased estimators in any case, to avoid introducing even such small biases. That is a good point which we should have indicated in the paper. The reason is that it leads to simpler expressions for the comparison of the standard and block estimators. So, we get an equation (10) in which one can easily interpret the difference between the block estimator and standard estimator in terms of an additional term that is the variance of the monthly averages. In effect, this result is just the well-known result expressed in terms of sums of squares in

ANOVA (e.g. see Kottegoda and Rosso (2008), p. 285). Since the reason for putting in this equation is just to give some understanding of where the differences between the estimators comes from, this seems sufficient for that purpose. Ultimately, significant differences between block and standard estimators only arise when dealing with ratios, as we explain in the paper, and for these, there are no useful such equations to write down.

In fact, the numerical estimates that we have shown are for the unbiased estimators, so that removes any concern about the impact of the biases in question. However, this points to the fact that we need to clarify these issues in the paper which, as currently presented, can be confusing. We therefore add some text to indicate that for the biased (but asymptotically unbiased) estimators of the variances, we get the relations shown in the equations, while the relations are a little more complex when using unbiased estimators. We also add a sentence to indicate that the numerical values are for unbiased estimators. We also indicate that this brief investigation is not aimed at making any general point about block estimators, but simply indicating the problems that arise in the case of the estimator of a ratio like the coefficient of skewness.

**References**

Kottegoda, N.T. and Rosso, R. (2008) *Applied Statistics for Civil and Environmental Engineers*, 2nd ed., London: Blackwell

Rodriguez-Iturbe, I., Cox, D.R., Isham, V. (1988) A point process model for rainfall: further developments*, Proc. Roy. Soc. Lond.*, A417, 283-298

---

## Author Response (AR2)

We thank Dr Nadav Peleg (the Editor) for his comments and for his coordination of the entire review process.

**(1) The Editor suggests to move Appendices to the Supplement.**

*Appendices. Please consider moving the appendices to Supplementary Material to reduce the length of the paper.*

**Ans:** Thank you for this suggestion. We would prefer to keep the Appendix A because it is highly relevant to the property derivation in the main context, but we agree to move Appendices B and C to the Supplement.

**(2) The Editor suggests to improve Figure 1.**

*Figure 1. Consider adding an arrow to indicate the timeline.*

**Ans:** Thank you for the suggestion. An arrow has been added to indicate the timeline.

**(3) The Editor suggests to reduce the number of digits after the decimal point in Tables 1-4 and other parts in the manuscript.**

*Tables 1-4 (and elsewhere in the text). Do you really need to present the figures with 4 digits after the decimal point? Consider reducing the digits after the decimal point to 2 or 3 (if needed).*

**Ans:** Thank you for the suggestion. We have reduced the number of digits after the decimal point to 3.

We thank Reviewer#1 for his careful reading of the paper and his perceptive comments.

Here are some responses to the points made by Reviewer#1.

**(1) Reviewer#1 suggests to further investigate the impact of parameter $\alpha$ to the statistical property of parameter $\eta$.**

*I agree that VAR(eta) = alpha/nu^2. I was mistaken. It might worth investigating how the mean and variance of the parameter eta behave for the case in which the parameter alpha is lower than 1 and for the case it is greater than 1 based on the 12 parameter sets that were already estimated for Bochum data. I believe this investigation will enlighten the strength of this model.*

**Ans:** I think there is a misunderstanding here. In the randomised models, $\eta$ is no longer a parameter (as it was in the non-randomised model). Rather, it is a random variable whose mean and variance depend upon parameter $\alpha$: there are analytical relations between the mean and variance and $\alpha$ (as in the equation referred to in the reviewer's comment; the other is $E[\eta] = \frac{\alpha}{\nu}$). So, presumably, what the reviewer is referring to as the variance of $\eta$, is something like the variability between the different parameter sets of the mean value of $\eta$, i.e. $E[\eta] = \frac{\alpha}{\nu}$, for each parameter set.

From that equation, it is clear that, assuming $\nu$ does not change, the values of $E[\eta] = \frac{\alpha}{\nu}$ would get larger when $\alpha$ increases so that the variability of $E[\eta]$ would also increase when $\alpha$ increases. But in fact, parameter $\nu$ also changes, so there is nothing than can be concluded a priori. The sample of 12 parameter sets is very small, so we fitted the model to a few more data sets and obtained the figure on the left below. For completeness sake, we include a similar evaluation of the variability of $Var[\eta]$ with $\alpha$.

[Figure]

[Figure]

This shows that, aside from three parameter sets for small values of $\alpha$, there is no visible trend in the variability of either $E[\eta]$ or $Var[\eta]$ with $\alpha$. This is confirmed by fitting a regression line to these points and observing that the gradient is not significantly different from 0 with 95% confidence (see also the small values of the coefficient of determination). We do not think that including either of these figures would add much to the paper; and since they make use of data that were not utilised in the paper, their inclusion would require first presenting these other data sets, which would significantly lengthen the paper. We therefore propose to leave out this analysis.

**(2)  Reviewer#1 points out the limitation of fixed-window annual maximum.**

*While I agree with your point (thus no additional works requested on this matter), but I am not too sure whether the fixed window approach is a standard (at least it is not standard in the US). If it is standard, I think the standard needs to be improved because the fixed-window annual maximum rainfall is not true annual maximum rainfall.*

**Ans:** Thank you for the explanation. As mentioned previously, we chose to use the fixed window approach, so our results could be easily compared with those from the previous relevant works (such as Kaczmarska, et al., 2014). We would consider using a moving window approach in our future work.

**(3)  Reviewer#1 confirms that the overestimation of proportion dry can be resolved by filtering out trivial rainfall or by narrowing down search space of phi and kappa.**

*I also confirmed that the overestimation of PD is caused by very trivial rainfall (e.g. 0.001mm) from my simulation. I also found that this problem can be fixed when you narrow down the search space of the parameter phi and kappa. You may also want to either briefly mention it or investigate it.*

**Ans:** Thank you for confirming that the overestimation issue can be resolved by filtering out trivial rainfall and for the suggestion on restricting the search space of phi and kappa. Since we have not carried out any investigation on restricting the search space of kappa and phi, we cannot make any claims in the paper about the usefulness of doing this. This is something that will have to be examined in a separate publication.

**References**

Kaczmarska, J., Isham, V., and Onof, C.: Point process models for fine-resolution rainfall, Hydrological Sciences Journal, 59, 1972–1991, https://doi.org/10.1080/02626667.2014.925558, 2014.